# LOCAL AUTOREGRESSION WITH FINITE-SUPPORT RANDOM VARIABLES FOR IMAGE GENERATION

## ABSTRACT

We propose the Finite-Support Local Autoregressive (FS-LAR) model, a novel approach based on finite support random variables that capture local pixel dependencies for image generation. Our approach adopts a Frequentist perspective. Instead of imposing priors on the target distribution, we make assumptions in the data processing procedure using an autoencoder. We observe that pixel dependencies are decoupled after reconstruction, despite negligible reconstruction error. In reconstructed images, pixel dependencies rely entirely on the latent representations and the decoder architecture. By designing the decoder architecture, we can control the range of pixel dependencies, which are then modeled by finite support random variables. The generation process performs global sampling based on random variables whose dependencies are controllable, enabling an exponential reorganization of local features in reconstructed images. Our approach has several interesting properties. Theoretically, we embrace the empirical distribution, eliminating the need to prevent overfitting. Since the support of the random variables is finite, it is possible to exhaustively search all possible generated images to verify its certifiability. Since no prior is imposed, the target distribution is explicitly known and can be fully characterized. Practically, the generation quality is promising compared to state-of-the-art methods, even without using a network in the generation process. Moreover, the proposed approach is able to perform generation with a limited number of images. Finally, the generated images are inherently interpretable, as they are reorganizations of locally independent pixels or patches.

## 1 INTRODUCTION

Although modern probabilistic generative models, including VAEs, GANs, and diffusion models rest on distinct theoretical assumptions, they seem to encounter the same fundamental theoretical tension: the trade-off between accurately fitting the empirical data distribution and avoiding overfitting through memorization (Fit–Overfit Tension). Most of these methods, as well as their extensions, aim to closely approximate the empirical training distribution to produce realistic samples (e.g., via reconstruction in VAEs). However, to handle the risk of memorization, they necessarily incorporate inductive biases or constraints, such as KL bottlenecks in VAEs, noise schedules in diffusion models, and

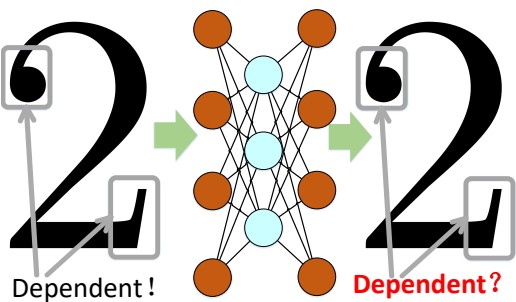

Figure 1: Although the reconstructed images are barely different from the original ones, their pixel dependencies have completely changed.

discriminator regularization in GANs. This inherent tension has become a central paradox in probabilistic generative modeling and led to several theoretical and practical issues. For example, theoretically, none of these methods can determine the true distribution of data without additional assumptions given a finite number of samples (Sec. A.1.1). Practically, controlling the balance between fitting and overfitting requires explicit inductive biases. In this paper, we propose a novel generative model with the potential to address this paradox.

The root of the paradox lies in the tension between epistemic uncertainty about the true distribution (as emphasized in the Bayesian view) and the practical reliance on the empirical distribution derived from finite data. Our idea is to embrace the empirical distribution following the Frequentist perspective [1]. Instead of imposing priors on the true data-generating distribution, we focus on the data-processing mechanism. Specifically, we assume that the pixel dependencies in the images reconstructed by the autoencoder differ from those in the original images. This assumption is widely held in almost all autoencoder architectures; because reconstructed images reflect the decoder's architectural dependency graph, which is not guaranteed to match the dependency structure of the original images. For example, as shown in Fig. 1, although the reconstructed images exhibit negligible reconstruction error, their pixels dependency structure has changed. Thus, it is possible to control the range of pixel dependencies through decoder architectures, and our Finite-Support Local Autoregressive model (FS-LAR) is proposed based on this insight.

Our approach is essentially a local autoregressive process based on finite support random variables that model local pixel dependencies. We first design an autoencoder whose decoder has controlled receptive fields to regularize the range of pixel dependencies, in order to constitute locally dependent latent representations. Then, by injecting noise with finite support (e.g., a Rademacher distribution), the latents are extended into locally dependent random variables whose joint distribution can be learned. Finally, the generation process is designed as a global sampling procedure based on the learned local joint distributions, subject to sliding windows for sampling. The generated images are reconstructed from the stitchings of sampled latents, which can be interpreted as reorganizations of pixels or patches with local dependencies. The number of these stitchings grows exponentially with the number of sampling window positions, thereby ensuring the diversity of the generated content. There are four important properties of our method:

- **Fidelity:** The locally dependent random variables used for reconstruction are the same as those used for global sampling. Patch consistency in the generated images is regularized by the reconstruction error, thereby ensuring fidelity (Sec. A.3.1).

- **Diversity:** By adjusting the noise distribution, we can control the joint distribution of locally dependent random variables. This, in turn, influences the probability of specific latent stitchings being sampled, thereby controlling diversity (Sec. A.3.2).

- **Interpretability:** The generated images can be interpreted as recombinations of pixels or patches constrained by local dependencies. Furthermore, the sampling distribution is identified as a mixture of distributions of locally dependent random variables, which is fully characterized. The autoregressive sampling process does not rely on neural networks (Sec. A.3.3).

- **Certifiability:** Because the locally dependent random variables have finite support, the number of possible global sampling outcomes is also finite. Theoretically, it is possible to exhaustively search all possible generated images to verify certifiability. Practically, we can draw a finite number of samples to construct an empirical distribution, which then replaces the original distribution for global sampling. In short, certifiability becomes a verifiable property rather than a heuristic (Sec. A.3.6).

## 2 PHILOSOPHICAL AND CONCEPTUAL BACKGROUND

This section provides the philosophical and conceptual background of our work, clarifying how differing views on probability inform generative modeling choices and evaluation, and setting up the fit–overfit tension that motivates our approach. In philosophy, Frequentists consider distributions as an objective entity, essentially defined as the long-term frequency of a particular event occurring over time. Hence, it is treated as something that exists independent of the observer, and assumptions imposed on it need to be minimized. In contrast, Bayesians regard distributions as a form of subjective probability, a quantification of our personal belief about the likelihood of an event. Because of this subjectivity, Bayesian methods are more flexible with respect to the prior assumptions on

---

[1]We adopt a philosophical definition of Bayesian vs. Frequentist approaches (De Finetti, 1974/2017), based on the degree of prior assumptions about the true distribution. For instance, autoregressive models make minimal assumptions and are thus closer to the Frequentist view, whereas VAEs impose Gaussian assumptions in the latent space, aligning more with the Bayesian perspective.

the distribution. A key difference between the two schools lies in their attitude toward empirical distribution. Under the Frequentist view, with finite data the empirical distribution is considered the best available approximation of the true distribution (Glivenko–Cantelli theorem (Vaart & Wellner, 1997)), whereas the Bayesian perspective does not endorse this conclusion. A similar conflict appears in modern generative models, as almost all generative models do not treat the empirical distribution as their final target. A model that perfectly fits the empirical distribution is considered as overfitting, a memorization model without generative ability. To avoid this, inductive biases or structural constraints are introduced to endow the model with generalization capacity. This reveals the inherent fit–overfit tension. However, from our perspective, the necessity of preventing overfitting no longer exists. As long as overfitting occurs locally, the generative model still has generative capability, which inspires us to propose the Finite-support local autoregressive model.

## 3 RELATED WORK

Probabilistic generative models have achieved remarkable progress recently (Bond-Taylor et al., 2022). In particular, Variational Autoencoders (VAEs) (Kingma & Welling, 2014), Generative Adversarial Networks (GANs) (Goodfellow et al., 2014), diffusion models (Song et al., 2020; Ho et al., 2020), and autoregressive models (Van Den Oord et al., 2016) are currently the most widely used approaches. Although the techniques behind these methods differ, almost all of them embody fit–overfit tension. Variational autoencoders assume a Gaussian prior in the latent space (Kingma & Welling, 2014), with a reconstruction loss designed to fit the training data and a KL term that regularizes the posterior, effectively mitigating overfitting. This fit–overfit tension or reconstruction–regularization trade-off is further controlled by a user-defined weight in $\beta$-VAE (Higgins et al., 2017), adjusting the balance between reconstruction fidelity and diversity (latent capacity). In contrast, Gaussian mixture variational autoencoders (GMVAEs) (Dilokthanakul et al., 2016; Yang et al., 2019; Guo et al., 2020) replace the standard prior with a mixture of Gaussians, to address the "prior hole" problem (Aneja et al., 2021; Xiao et al., 2020; Nalisnick et al., 2018). While this increases the flexibility of the latent distribution for better fitting, it tends to reduce generative diversity. Diffusion models also exhibit a similar fit–overfit tension (Sohl-Dickstein et al., 2015; Song et al., 2020; Ho et al., 2020; Song et al., 2021), but in a more concealed form. Although the prediction targets of different diffusion formulations differ, namely the original image (Sohl-Dickstein et al., 2015), the added noise (Ho et al., 2020), or the score function (Song et al., 2020), they can all be considered "fit" objectives that encourage sampled images to approach the training data. In contrast, overfitting is implicitly mitigated by the nature of the inputs, which are noisy versions of the training images (Yang et al., 2023). The injected noise induces a one-to-many mapping non-convergence (Sec. A.1.5) or a stochastic reverse process, so a fixed noisy input can correspond to multiple plausible reconstructions, capturing uncertainty (Feng et al., 2023). This helps prevent simple memorization of training data (Feng et al., 2023). Note that sampling typically starts from pure Gaussian noise. Nevertheless, diffusion models can still reproduce training samples under certain conditions (Carlini et al., 2023; Somepalli et al., 2023). The fit-overfit tension in Generative Adversarial Networks (GANs) (Goodfellow et al., 2014) is quite unique. First, the input of a generator is usually samples from pure noise, which makes one-to-many non-convergence in GANs more severe compared to diffusion models. Besides, the generator in GANs is optimized with the gradient from a discriminator rather than training samples (Huang et al., 2024). Both these factors prevent GANs from overfitting, which could be the reason why GANs have the least possibility of generating training samples (Akbar et al., 2025; Carlini et al., 2023).

Autoregressive models have the most subtle fit–overfit tension (Bengio et al., 2003; Vaswani et al., 2017; Wang et al., 2025; Bachmann et al.; Yu et al., 2025). The "fit" part is easy to see, since an autoregressive model is basically a factorization of the joint distribution by the chain rule. The overfit mitigation part exists practically rather than theoretically in many autoregressive models (Li et al., 2024; van den Oord et al., 2016). Theoretically, the generation of token $\mathbf{z}_N$ is based on all the previous tokens $\mathbf{z}_{<N}$. Based on the joint distribution factorization formula in the work of van den Oord et al. (2016), an autoregressive model's target effectively aligns with the empirical distribution of the training data, with finite data and enough learning ability. This helps explain why autoregressive models are particularly prone to memorization (Carlini et al., 2021; Nasr et al., 2023). However, the observation field of an autoregressive model can have difficulty covering all previous tokens practically. This is especially true for data with infinite length like language data, which can keep continuing because we can keep writing (Shannon, 1948). It is actually the locality that endows the

autoregressive model with generation ability, which inspired us to utilize locality to guarantee the generation ability of the autoregressive model (Zhao & Basu, 2025). For data with a clear length like images, the autoregressive model is more likely to generate training images (Kowalczuk et al., 2025). There are a few autoregressive models that incorporate local patterns (Mao et al., 2024; Cao et al., 2021) or mask (Chang et al., 2022; Yu et al., 2025), but most of them are proposed to reduce computational complexity, rather than realize that the generation ability comes from locality. For example, Cao et al. (Cao et al., 2021) proposed a Local Autoregressive Transformer that restricts attention regions to accelerate inference. In contrast, Zhao & Basu (2025) suggested that the autoregressive model falls into memorization when learning with the global distribution, and that the autoregressive model should instead focus on the local distribution. Unfortunately, they did not realize that locality already exists in practical network architectures. In the proposed approach, we make assumptions in the data processing procedure with an autoencoder architecture to control the dependencies. More importantly, the tokens in the proposed approach are not deterministic values but random variables with finite support. The controllable local dependencies (Locality) and latent random variables (Stochasticity) are the two key differences between the proposed approach and previous autoregressive models for image generation (van den Oord et al., 2016; Zhao & Basu, 2025; Cao et al., 2021). Another key difference is that our approach also does not require a neural network during the sampling procedure.

## 4 FINITE-SUPPORT LOCAL AUTOREGRESSIVE MODEL

### 4.1 LOCAL DEPENDENCIES MODELING

The proposed approach is essentially an autoregressive model based on locally dependent latent random variables with finite support, which emphasizes the locality and stochasticity (Sec. A.1.3). Mathematically:

$$p(\mathbf{Z}) = \prod_{i=1}^{M} p(\mathbf{z}_i | \mathbf{z}_{[i-\rho,i)}) = \prod_{i=1}^{M} \sum_{n=1}^{N} p(\mathbf{z}_i \mid \underbrace{\tilde{\mathbf{G}}_n}_{\text{stochasticity}}) p(\tilde{\mathbf{G}}_n \mid \underbrace{\mathbf{z}_{[i-\rho,i)}}_{\text{locality}}), \tag{1}$$

where $\mathbf{Z} = \{\mathbf{z}_i\}_{i=1}^{M}$ denotes the latent representation of images, with $\mathbf{z}_i$ as its entries. $\rho$ is the window size describing locality. $\tilde{\mathbf{G}}_n = \{\tilde{\mathbf{z}}_j \mid j \in [0, 2\rho + 1]\}$ is a patch of locally dependent random variables $\tilde{\mathbf{z}}_j$, extended from latents by adding noise sampled from a specific distribution, i.e., $\tilde{\mathbf{z}}_j = \mathbf{z}_j + \tau \cdot \epsilon$, where $\epsilon$ denotes the noise and $\tau$ controls its intensity. Note that the center of $\tilde{\mathbf{G}}_n$ maps to the position of the next token $\mathbf{z}_i$ waiting for inference. The key to enabling this autoregressive process is to construct locally dependent random variables. We achieve this by designing the decoder so that, for each center $i$, the latent patch $\tilde{\mathbf{G}}$ covers the receptive field of the pixels it reconstructs. Consequently, any decoded pixel (or patch) depends only on the variables within $\tilde{\mathbf{G}}$, thereby controlling the local dependencies between the latents and the reconstructed pixels. A typical architecture of our decoder is demonstrated in Tab. 1.

We next discuss the training of our autoencoder. Note that locality is not explicitly shown in the mathematical expression, but is enforced through the design of the decoder architecture (Sec. A.4). Mathematically:

$$\mathcal{L}(\mathbb{X}; \theta, \phi) = \sum_{s} \|d_\theta(\tilde{\mathbf{Z}}^{(s)}) - \mathbf{X}^{(s)}\|^2 + \beta(t) \cdot \sum_{s} \|\tilde{\mathbf{Z}}^{(s)}\|^2, \quad \tilde{\mathbf{Z}}^{(s)} = \tanh\big(e_\phi(\mathbf{X}^{(s)})\big) + \tau \cdot \epsilon, \tag{2}$$

$e_\phi(\cdot)$ and $d_\theta(\cdot)$ denote the encoder and decoder, with $\phi$ and $\theta$ as their parameters. $\beta(t)$ is a monotonically decreasing schedule that anneals from $\beta_0$ to 0 during training. $\mathbb{X} = \{\mathbf{X}^{(s)} \mid s = 1, 2, \cdots, S\}$ is the set of training images $\mathbf{X}^{(s)}$. $\tilde{\mathbf{Z}}$ is the latent representation of an image $\mathbf{X}$, whose entries $\tilde{\mathbf{z}}_i$ are random variables with local dependencies controlled by the architecture of the decoder. $\tilde{\mathbf{G}}_n$ are patches captured from $\tilde{\mathbf{Z}}$, where the patch size is equal to or greater than the receptive field of the decoded pixels or patches. $\epsilon$ is noise sampled from the generalized triangular distribution $\epsilon \sim \text{Tri}(\kappa)$, where $\kappa$ controls the sharpness of the peak:

$$\epsilon \sim \text{Tri}(\kappa) = \begin{cases} (1 - u^\kappa), & \text{if } u > 0^+ \\ (|u|^\kappa - 1), & \text{if } u < 0^-, \end{cases} \tag{3}$$

where $u \sim \mathcal{U}(-1, 1)$ denotes the uniform distribution over $[-1, 1]$. When $\kappa = 1$, $\epsilon$ becomes uniform noise. Although the random variables $\tilde{\mathbf{z}}_i$ are locally dependent, their support is still not finite, because there are infinitely many real numbers within a finite support range. In order to restrict the support of the random variables, we employ an indicator function to constrain the support of the latent random variables after the autoencoder in Eq. 2 is well trained. Then, an additional training phase of the decoder is used to adapt to inputs restricted to $\pm 1$. Mathematically:

$$\mathcal{L}(\mathbb{X}; \theta) = \sum_s \left\| d_\theta \big( 2 \cdot \mathbf{1} \left[ \tilde{\mathbf{Z}}^{(s)} > 0 \right] - 1 \big) - \mathbf{X}^{(s)} \right\|^2. \tag{4}$$

The random variables $\tilde{\mathbf{z}}_i$ then become random variables $\tilde{\mathbf{z}}_i^{\pm}$ following a Rademacher distribution with finite support $\{-1, 1\}^{\|\tilde{\mathbf{z}}_i^{\pm}\|_0}$. Mathematically:

$$\tilde{\mathbf{z}}_i^{\pm} = 2 \cdot \mathbf{1}[\tilde{\mathbf{z}}_i > 0] - 1 \sim \mathbb{B}_\pi^{\|\mathbf{z}\|_0}(\mathbf{z}; \boldsymbol{\mu} = \mathbf{z}_i) := P(\mathbf{z} = \mathbf{k}) = (1-\pi)^{\|\boldsymbol{\mu} \oplus \mathbf{k}\|_1} \cdot \pi^{\|\boldsymbol{\mu}\|_0 - \|\boldsymbol{\mu} \oplus \mathbf{k}\|_1}, \tag{5}$$

where $\oplus$ denotes the Hamming distance. The probability value of $\pi$ is related to the parameters $\tau$ and $\kappa$. When $\tau > 1$, we have $\pi = 1 - 0.5 \cdot (1 - \frac{1}{\tau})^\kappa$. When $\tau \leq 1$, we have $\pi = 1$, and $\tilde{\mathbf{z}}_i^{\pm}$ degenerates into deterministic variables (Sec. A.2.3).

The autoencoder in Eq. 2 is closely related to the $\beta$-VAE, but it uses a $\tanh$ activation to restrict the values of the latent variables output by the encoder. Given the exclusivity in the overlap of the probability spaces of different random variables, the latent representation tends to approach the boundary values of $\tanh$ (Sec. A.1.7). Thus, the encoder output $\mathbf{z}_i$ gradually converges to signed binary values $\pm 1$. After this, the indicator function can be applied to $\tilde{\mathbf{z}}_i$ to create random variables $\tilde{\mathbf{z}}_i^{\pm}$, which are used for global sampling in the generation process. Since the encoder output becomes signed binary after the autoencoder is well trained, applying the indicator function does not have a significant effect on the reconstruction loss (Fig. 9). Consequently, the true distribution that produces samples for generating images is a mixture of the Rademacher distributions from these latent random variables $\sum_i p(\tilde{\mathbf{z}}_i^{\pm})$. In fact, the mixture of distributions from latents raises an interesting logical inconsistency regarding the common pre-assumption in VAEs, the sampling distribution (a single Gaussian) differs from the effective distribution (a mixture of Gaussians) used in reconstruction (Sec. A.1.4). In contrast, in the proposed approach, the distribution used for reconstruction is also the one used for global sampling, which aligns with a Frequentist perspective that avoids making prior assumptions about the true distribution, with the motivation of addressing the fit–overfit tension.

## 4.2 GLOBAL SAMPLING GENERATION

The generation process is a global sampling based on random variables $\tilde{\mathbf{z}}^{\pm}$, which are processed by the indicator function. Mathematically:

$$p(\mathbf{Z}^{\pm}) = \prod_{i=1}^M p(\mathbf{z}_i^{\pm} | \mathbf{z}_{[i-\rho, i)}^{\pm}) = \prod_{i=1}^M \sum_{n=1}^N p(\mathbf{z}_i^{\pm} | \tilde{\mathbf{G}}_n^{\pm}) p(\tilde{\mathbf{G}}_n^{\pm} | \mathbf{z}_{[i-\rho, i)}^{\pm}). \tag{6}$$

Since the probability mass function of latent random variables $\tilde{\mathbf{z}}^{\pm}$ is fully characterized in Eq. 5, the conditional distribution terms in Eq. 6 can be rewritten as (Sec. A.2.1):

$$p(\mathbf{z}_i^{\pm} | \tilde{\mathbf{G}}_n^{\pm}) = P(\mathbf{z}_i^{\pm} = \mathbf{k} | \text{Mid}(\tilde{\mathbf{G}}_n^{\pm}) = \mathbf{c}_n) = (1-\pi)^{\|\mathbf{k} \oplus \mathbf{c}_n\|_1} \cdot \pi^{\|\mathbf{k}\|_0 - \|\mathbf{k} \oplus \mathbf{c}_n\|_1}$$

$$p(\tilde{\mathbf{G}}_n^{\pm} | \mathbf{z}_{[i-\rho, i)}^{\pm}) = P(\text{Pre}(\tilde{\mathbf{G}}_n^{\pm}) = \mathbf{g}_n \mid \mathbf{z}_{(i, i-\rho]}^{\pm} = \mathbf{z}^\circ) = \frac{(1-\pi)^{\|\mathbf{g}_n \oplus \mathbf{z}^\circ\|_1} \cdot \pi^{\|\mathbf{z}^\circ\|_0 - \|\mathbf{g}_n \oplus \mathbf{z}^\circ\|_1}}{N \cdot P(\mathbf{z}^\circ)}, \tag{7}$$

where $\text{Mid}(\cdot)$ and $\text{Pre}(\cdot)$ denote the operations to extract the middle and previous parts of $\tilde{\mathbf{G}}_n$, respectively, $\mathbf{c}_n$ and $\mathbf{g}_n$ are their specific observation values. We have $\|\mathbf{g}_n\|_0 = \|\mathbf{z}^\circ\|_0$ and $\|\mathbf{c}_n\|_0 = \|\mathbf{z}_i^{\pm}\|_0$. $N$ is the total number of $\tilde{\mathbf{G}}_n^{\pm}$, and $P(\mathbf{z}^\circ)$ is a constant value. Both $N$ and $P(\mathbf{z}^\circ)$ cancel out after probability normalization. By using Eq. 7 and Eq. 6, the probability for sampling can be directly computed. The true distribution used for sampling is thus fully characterized, consistent with one of the central claims in this work. Consequently, it becomes optional to either use a neural network to approximate this distribution or to perform direct sampling without a neural network. To demonstrate this, we propose three different sampling strategies, along with discussions of their advantages and disadvantages.

**Direct sampling:** Direct sampling refers to using Eq. 7 to compute the probability given the observed value directly. By plugging Eq. 7 into Eq. 6, we obtain (Sec. A.2.2):

$$P(\mathbf{k}|\mathbf{z}^\circ) = \frac{\sum_{n=1}^N \exp\big(\eta(\mathbf{k}; \tilde{\mathbf{G}}_n^\pm, \mathbf{z}^\circ) \cdot \log(1-\pi) + (\lambda - \eta(\mathbf{k}; \tilde{\mathbf{G}}_n^\pm, \mathbf{z}^\circ)) \cdot \log\pi\big)}{\sum_{\mathbf{k}} \sum_{n=1}^N \exp\big(\eta(\mathbf{k}; \tilde{\mathbf{G}}_n^\pm, \mathbf{z}^\circ) \cdot \log(1-\pi) + (\lambda - \eta(\mathbf{k}; \tilde{\mathbf{G}}_n^\pm, \mathbf{z}^\circ)) \cdot \log\pi\big)}, \quad (8)$$

where $\lambda = ||\mathbf{k}||_0 + ||\tilde{\mathbf{z}}^\circ||_0$ and $\eta(\mathbf{k}; \tilde{\mathbf{G}}_n^\pm, \mathbf{z}^\circ) = ||\mathbf{k} \oplus \mathbf{c}_n||_1 + ||\mathbf{g}_n \oplus \mathbf{z}^\circ||_1$. Both $\lambda$ and $\eta(\cdot)$ are computationally convenient in practice. Direct sampling offers strong interpretability and fidelity, while also allowing for controllable diversity. By adjusting the parameters of the generalized triangular distribution, we can control the probability mass function, which in turn affects the reconstruction quality. The main disadvantage is that all random variable patches need to be involved during sampling, which leads to increased computational cost. However, thanks to the parallel computation capability of GPUs, this additional cost remains acceptable. We regard this as the most effective sampling strategy under our limited computational resources, and most of the images, including those compared with state-of-the-art methods, are generated using this approach (Sec. A.3.7).

**Exhaustive searching:** Although the support of the sampling distribution is finite, the number of possible values grows rapidly with the increasing number of dimensions. One way to handle this is by drawing a finite number of samples to construct an empirical distribution, which then replaces the original distribution for global sampling. These samples are fed into the decoder for further training, ensuring that the decoder has seen each of them. Subsequently, the same samples are employed together with exhaustive search algorithms. This method provides the safest approach for generation, and the quality of the generated results is also the highest. Meanwhile, since every possible generated image can be certified, the proposed approach is suitable for sensitive fields, such as medical imaging. Unfortunately, its diversity is quite limited, since the number of samples drawn is usually much smaller than the full support in high-dimensional spaces (Sec. A.3.6).

**Network approximation:** The third approach employs a neural network to approximate the sampling distribution. This is essentially a network that takes the prior tokens as input to predict the next token, as shown in Eq. 6. The key advantage of this approach is flexibility, as the autoregressive network is the only component that needs to be retained once the model is well trained. Its primary limitation, however, lies in the lack of interpretability, since a neural network is involved in the sampling process and there is no guarantee that the outputs remain within the original distribution. Another drawback is the computational cost. Without sufficient parameters, it is difficult to train a network that effectively learns the sampling distribution (Sec. A.3.3).

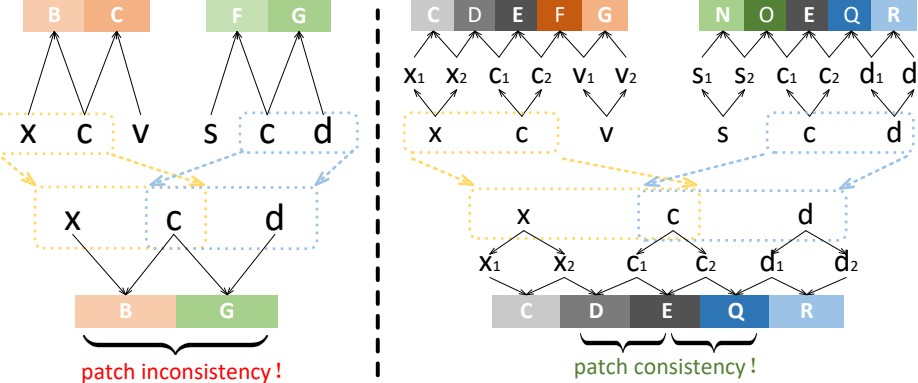

Figure 2: Illustration of patch inconsistency and patch consistency. By adding a transposed convolution before reconstruction, patch consistency in the generated images can be ensured.

An important concern is patch consistency. As illustrated in Fig. 2, directly applying the receptive field as the dependency range $\rho$ guarantees the fidelity of reconstructed patches, but does not ensure consistency across patches. To address this, several architectural strategies can be adopted. One convenient approach is to extend the observation range beyond the receptive field, for example by including one or more additional positions. An alternative approach is to apply a transposed convolution (ConvT) followed by a standard convolution, which naturally introduces a transition where intermediate pixels act as consistency-preserving boundaries. This strategy is employed in the demonstration experiment on patch consistency (Sec. A.3.1).

# 5 EXPERIMENTS

## 5.1 ABLATION STUDY ON MEMORIZATION

Since the proposed approach embraces the empirical distribution, a potential concern is that many generated images may closely resemble the training data. For evaluation, we propose the Unique Image Rate (UIR) defined as the proportion of generated images that differ substantially from the training set. The computation of UIR is straightforward. For every generated image, we first identify its closest training image and compute the corresponding Peak Signal-to-Noise Ratio (PSNR) [2]. Then, a PSNR threshold $\gamma$, usually set to 25 or 30, is applied to exclude generated images that are visually similar to training images. Finally, the UIR is obtained as the ratio between the number of images that pass this threshold and the total number of generated images. Mathematically:

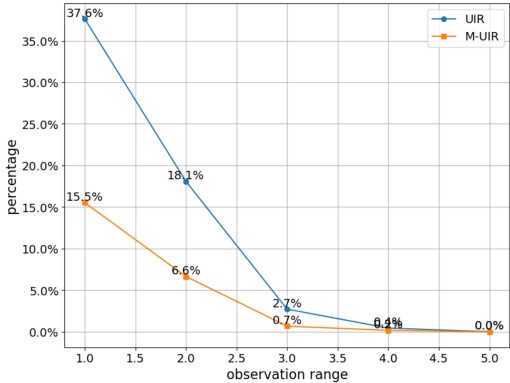

Figure 3: Demonstration of UIR and M-UIR with respect to the observation range. As the observation range increases, more and more generated images resemble the training data.

$$\text{UIR} = \frac{\sum_n \mathbf{1}\left[\text{PSNR}(\mathbf{X}^{(n)}, \hat{\mathbf{X}}^{(n)}) < \gamma\right]}{N}, \tag{9}$$

where $\mathbf{X}^{(n)}$ denotes the generated image and $\hat{\mathbf{X}}^{(n)}$ denotes the corresponding closest training image. $N$ is the total number of generated images. After filtering by a PSNR threshold, the remaining images are unique with respect to their closest training images. However, it is possible that these remaining images resemble each other. To address this issue, the PSNR filter is applied again between these remaining images to ensure that the PSNR values for every pair of remaining images are below the threshold $\gamma$. The ratio between the number of images after mutual filtering and the total number of generated images is then defined as the Mutual Unique Image Rate (M-UIR). Mathematically:

$$\text{M-UIR} = \frac{\left|\left\{\mathbf{X}^{(i)} \mid \forall \mathbf{X}^{(i)} \neq \mathbf{X}^{(j)}, \text{PSNR}(\mathbf{X}^{(i)}, \mathbf{X}^{(j)}) < \gamma\right\}\right|}{N}. \tag{10}$$

$|\{\cdot\}|$ denotes the operation to get the total number of elements in a set. We then utilize UIR and M-UIR to conduct an ablation study on memorization in the proposed approach with respect to the observation range. In particular, we use 500 images from CIFAR-10 for training and generation, with the architecture described in Tab. 1. The latent size is set to $8 \times 8$. With a higher observation range, the proposed approach becomes closer to a memorization model. Consequently, the UIR decreases accordingly. The relationship between observation range and UIR/M-UIR is shown in Fig. 3. In particular, we generate 5000 images to compute UIR and M-UIR. As shown in Fig. 3, when the observation range increases to 5, both UIR and M-UIR decrease to 0. This supports our argument regarding the empirical distribution. That is, when the global distribution is learned and the model has sufficient capacity (Sec. A.1.6, Sec. A.3.3), autoregressive models tend to memorize the empirical distribution.

## 5.2 EVALUATION OF THE PROPOSED APPROACH

We evaluate the properties of the proposed approach on several datasets, including MNIST (LeCun, 1998), CIFAR-10 (Krizhevsky et al., 2009), and CelebA (Liu et al., 2015). Given page limitations, most of the experiments are provided in the Appendix, including patch consistency (Sec. A.3.1), controllable diversity (Sec. A.3.2), image generation with 64 training images (Sec. A.3.4), High-resolution image generation (Sec. A.3.5), certifiability discussion (Sec. A.3.6), and the qualitative evaluation of the proposed approach on all three datasets (Sec. A.3.7). The source code will be public with the acceptance of this paper.

---

[2]PSNR can be replaced by other image distance metrics such as the Structural Similarity Index Measure (SSIM), we simply chose PSNR because it is easy to compute.

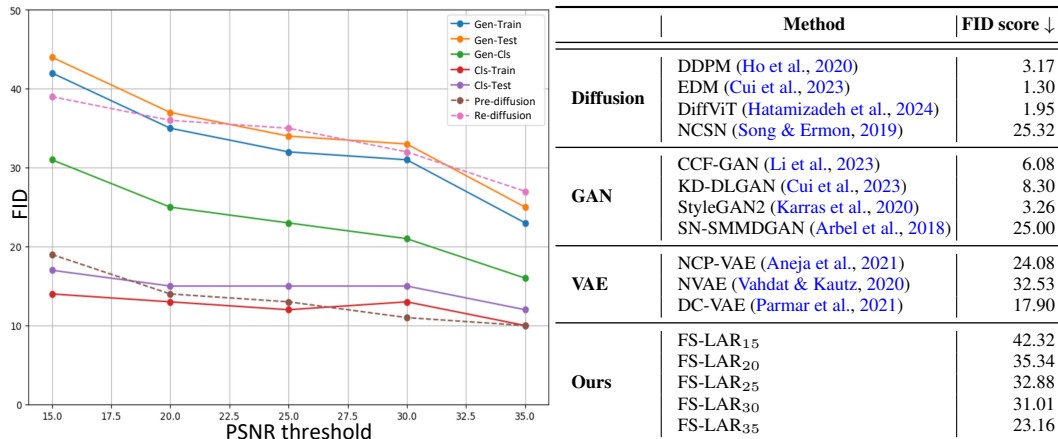

Figure 4: Left: The FID-PSNR curves of Gen-Train, Gen-Test, Gen-Cls, Cls-Train, Cls-Test, Pre-diffusion and Re-diffusion, after the generated images are filtered by PSNR values of 15, 20, 25, 30, and 35. Right: Comparison between the proposed approach and state-of-the-art methods on CIFAR-10 using FID.

Given our limited computational resources, the comparison with state-of-the-art methods is based on the CIFAR-10 dataset. For comparison, we utilize the autoencoder in Tab. 4, with a latent size of $7 \times 32 \times 32$ and $\rho = 5$. The FID value we obtained is 2.1, outperforming DDPM (Ho et al., 2020). However, considering the low UIR value of the proposed approach, such a low FID may result from the similarity between generated images and training images. To address this issue, we apply the M-UIR procedure to extract unique generated images using PSNR filter thresholds of 35, 30, 25, 20, and 15, and compute the FID scores of FS-LAR$_{35}$, FS-LAR$_{30}$, FS-LAR$_{25}$, FS-LAR$_{20}$, and FS-LAR$_{15}$, respectively. For example, an FID of FS-LAR$_{25}$ indicates that generated images whose PSNR (with respect to the closest training image) is greater than 25 are removed before computing the FID score, and, additionally, any two generated images must have a PSNR smaller than 25 between them. Consequently, every generated image used to compute FID is unique under a specific PSNR threshold. Comparison results are shown in the right part of Fig. 4. Although the proposed approach does not achieve SOTA-level FID (below 10, as in DDPM (Ho et al., 2020) or EDM (Cui et al., 2023)), the FID of FS-LAR$_{35}$ still outperforms NCSN (Song & Ermon, 2019), a milestone work in diffusion models. Moreover, the FID of FS-LAR$_{25}$ is close to NCSN (Song & Ermon, 2019) and better than NVAE (Vahdat & Kautz, 2020). Notably, the generated images filtered with a PSNR threshold of 25 are substantially different from each other, with a qualitative demonstration shown in Fig. 4.

**Discussion:** The fairness of comparisons with the proposed approach requires consideration of several factors. First, the autoencoder we utilized is relatively ordinary (Sec. A.4) and has a limited number of parameters. However, we cannot directly utilize good existing architectures, as their decoders do not satisfy locality. Second, none of the compared methods applied a PSNR filter during their FID computation. This makes it possible for some of their generated images to resemble training images. In addition, the computational cost of training is also an important factor (perhaps the most important factor), given our restricted computational resources (one RTX 4090 GPU with 24GB of memory). To investigate this, we downloaded a widely used DDPM implementation from GitHub and applied the same PSNR filtering strategy with different thresholds to evaluate its generated images. We report two sets of results, the Pre-Diffusion and Re-Diffusion, in the left part of Fig. 4. Pre-Diffusion is based on the publicly available pre-trained model, which achieved an FID of around 10. This differs from the 3.17 reported in Ho et al. (2020), possibly due to differences in network architecture or insufficient training epochs. Re-Diffusion is trained by us under the same computational budget as our autoencoder (12 hours on a single RTX 4090), to demonstrate the comparison under consistent evaluation settings of computational cost. The FID of Re-Diffusion is 32, which demonstrates that the proposed approach achieves competitive performance relative to DDPM (Ho et al., 2020), under consistent evaluation settings, particularly when controlling the computational budget and applying PSNR filtering.

**Objectivity:** The last important factor is related to the objectivity of the FID metric. FID is computed based on the assumption that the global distribution of generated images is similar to that of

Generated Images — Difference Map — Closest Training images

Figure 5: Images generated by the proposed approach trained on the CIFAR-10 dataset, with an FID score of 32 and a PSNR filtering threshold of 25.

the training images. However, the proposed approach generates images based on the distribution of locally dependent latents. As a result, the global distribution obtained from local sampling naturally differs from the global distribution of the training images. Thus, the lower bound of the FID for the proposed approach is inherently higher than that of previous methods. This raises another concern regarding the objectivity of FID comparisons between previous methods based on different prior assumptions, since methods whose priors are closer to the empirical distribution tend to achieve better FID scores (Sec. A.1.2). To demonstrate this, we compute the FID for Gen-Train, Gen-Test, Gen-Cls, Cls-Train, and Cls-Test. Specifically, "Gen" denotes the generated images, "Train" refers to the entire training set, "Test" refers to the testing set (unseen during training), and "Cls" denotes the set of closest training images with respect to the generated images. For example, Gen-Train means the FID is computed between generated and training images. As shown in the left part of Fig. 4, the FID of Gen-Cls is close to that of Cls-Train, which is the lower bound FID of the proposed approach. If the FID score obtained between generated and real images is consistently lower than the score obtained between two subsets of real images (e.g., FID of Cls-Train), the ability of FID to evaluate image realism becomes questionable. However, considering that FID (and its extensions) is currently one of the most popular quantitative evaluations we have, a more reasonable strategy is to use FID as a reference metric, but not as the sole criterion for judging whether one generative model is better than another.

**Interpretation and Limitation:** As shown in Fig. 5, the generated images of the proposed approach are essentially a reorganization of patches with high patch consistency, which results from global sampling based on locally dependent random variables. This validates the interpretability of the proposed approach. Unfortunately, although the patch consistency is pretty good, there are many images that violate semantic consistency. This is because the way we construct locally dependent random variables may not be ideally suited for images, which have a hierarchical architecture. Therefore, a multi-scale series of random variables would be more suitable for image data. Nevertheless, considering the workload of implementation, we reserve the multi-scale extension for future work. Another potential research direction focuses on the evaluation metric. Since the fidelity of the generated images in the proposed approach comes from patch consistency, one possible evaluation method could be the comparison between the patch consistency of the generated images and the training images.

## 6 CONCLUSION

We proposed the Finite-Support Local Autoregressive (FS-LAR) model for image generation, motivated by the Fit-Overfit tension. The proposed approach adopts a Frequentist perspective to make an assumption on the data process procedure rather than the "true" data distribution. In particular, we control the pixel dependencies in reconstructed images with the decoder architecture to construct locally dependent random variables with finite support. The generation process was designed as a global sampling based on these locally dependent random variables. Compared to previous generative models, fidelity was ensured by the reconstruction quality in an autoencoder. Diversity was controlled by adjusting the distribution of sampling random variables. Moreover, as no network was used during generation, the model is completely interpretable. Finally, since the support of random variables is finite, certifiability becomes a verifiable property rather than a heuristic.

## 7 ETHICS STATEMENT

This work does not present any ethical concerns. The datasets used are publicly available and contain no sensitive information.

## 8 REPRODUCIBILITY STATEMENT

We have made efforts to ensure the reproducibility of our work. The network architectures utilized in this paper are provided in Tab. 1, Tab. 2, Tab. 3 and Tab. 4. Details of the experimental setup are described in Sec. A.4. Proofs of the mathematical derivations are presented in Sec. A.2. Source code and running scripts will be released upon acceptance of this paper.

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

# A APPENDIX

## A.1 DISCUSSION OF STATEMENTS

### A.1.1 TRUE DISTRIBUTION IN GENERATIVE MODELS

In practice, the data we capture is always finite. Therefore, from a Frequentist's view, the empirical distribution is the best approximation of the true distribution—or equivalently, the true distribution in our real world. However, such a conclusion is generally unacceptable in modern generative modeling, since the purpose of a generative model is to produce new content. Accepting the empirical distribution as the true distribution would turn generative models into memorization models, which is considered overfitting[3]. Therefore, almost all previous generative models assume the existence of a "true" data distribution, which differs from the empirical distribution, usually based on the Manifold Hypothesis (Narayanan & Mitter, 2010; Fefferman et al., 2016). Although the exact form of this distribution is unknown, it is assumed to be capable of generating new samples beyond those observed in the training data. For example, VAEs optimize the evidence lower bound (ELBO) to minimize the divergence between the learned distribution and the true data distribution. GANs use an adversarial process to approximate the true distribution, while diffusion models learn the gradient of a distribution to gradually transform a Gaussian into the true image distribution. Unfortunately, there is still no explicit answer regarding what this true distribution actually is. Therefore, it is fair to conclude that previous approaches cannot uniquely identify the true data-generating distribution from finite samples without additional assumptions, as long as they do not accept the empirical distribution.

### A.1.2 COMPARISON OBJECTIVITY

An important issue arising from the fit–overfit tension is the objectivity of comparisons between different generative models. Currently, the Fréchet Inception Distance (FID) is the most widely used metric for evaluation. However, the fairness of such comparisons is questionable, as different models are based on different prior assumptions. In particular, models whose assumptions are closer to the empirical distribution tend to achieve better FID scores. For example, diffusion models are directly optimized with respect to training images, whereas GANs are optimized through an adversarial discriminator. It is therefore reasonable that diffusion models are closer to the empirical distribution compared with GANs. Consequently, it is unfair to simply claim that diffusion is superior to GANs solely because of better FID results (Dhariwal & Nichol, 2021). Similarly, it is also problematic to claim that autoregressive models are superior to diffusion models (Sun et al., 2024); this may simply be due to subjective biases in the evaluation introduced by different prior assumptions. Our main purpose is to propose a generative model that does not exhibit the fit–overfit tension by embracing the empirical distribution, which is typically considered overfitting. This is the main reason we claim that the proposed approach adopts a Frequentist perspective. Note that such overfitting occurs only on patches of pixels with local dependencies, which preserves generative ability.

### A.1.3 STOCHASTICITY AND LOCALITY

Unlike conventional autoregressive models, the proposed approach emphasizes locality and stochasticity. Locality is used to guarantee the model's ability to generate new content. Without locality, an autoregressive model without regularization tends to degenerate into a memorization model (Sec. A.1.6). Stochasticity relaxes the discrete latent representations. By turning latents into random variables, unmatched discrete tokens can be assigned a nonzero probability, which is very important in finite-sample settings, especially with limited data. For example, assume we have only two training sequences $(-1, 1, -1)$ and $(1, 1, 1)$, which provide two context–target pairs: $(-1, 1) \rightarrow -1$ and $(1, 1) \rightarrow 1$. In this setting, the unseen context $(1, -1)$ may appear even though it never occurs in the training instances. In contrast, such an unseen context can be assigned a nonzero probability through the random variables, enabling it to be matched probabilistically even if it never appeared in the training data. Mathematically:

$$(-1, \mathbf{1}, \mathbf{-1}) \xrightarrow[\text{No match:}(-1,1)\rightarrow -1]{\text{No match:}(1,1)\rightarrow 1} ? \quad \Rightarrow \quad (-1, \mathbf{1}, \mathbf{-1}) \xrightarrow[P(\mathbb{B}^2_\pi([1,1])=[1,-1])>0]{P(\mathbb{B}^2_\pi([-1,1])=[1,-1])>0} -1 \text{ or } 1. \quad (11)$$

---

[3]Note that any smoothing or post-processing (e.g., kernel density estimation) changes the empirical distribution; our discussion concerns the exact empirical distribution under finite samples.

Practically, when a network is used to approximate this autoregressive process, a standard neural network would still produce an output for such an out-of-support or out-of-distribution context. But this output can be unstable with limited training instances. In the proposed approach, the latent representations are extended into random variables and used to train the autoencoder. This means those unmatched patches effectively occur during training with some probability, which makes the probability computation stable and correct. Thereby, the proposed approach is able to perform generation with a limited number of training images (Sec. A.3.4).

### A.1.4 LOGICAL INCONSISTENCY IN VAE

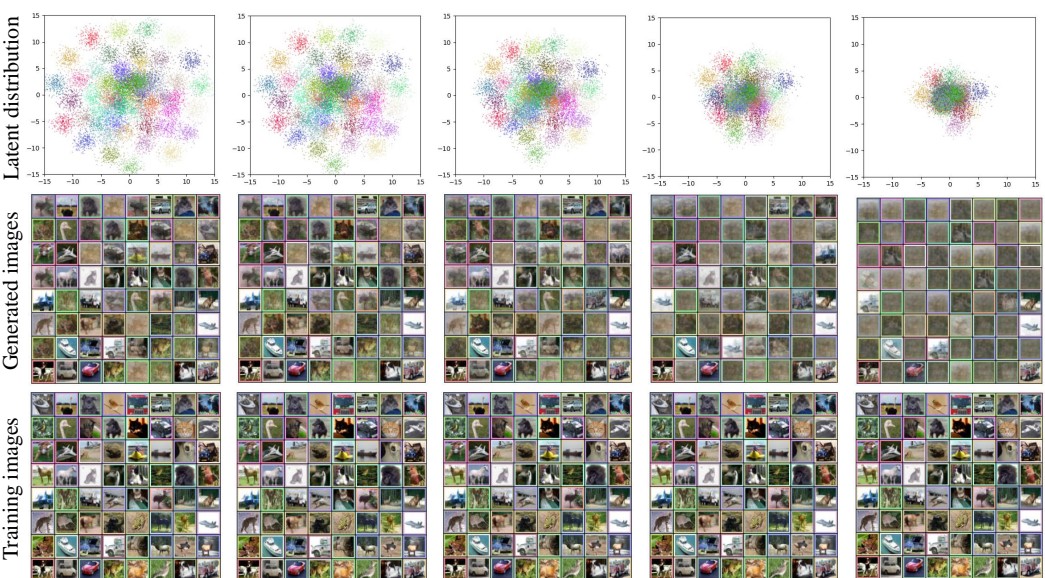

Figure 6: The VAEs trained with different weights of KL loss. From left to right, the weight of the KL loss is successively multiplied by 2, 4, 8, 16, and 32. As the weight of the KL loss increases, the latent random variables converge, ultimately approximating a standard Gaussian distribution. However, the quality of generation deteriorates as the overlap coefficient increases, leading to an increase in reconstruction loss.

An interesting logical inconsistency related to the true distribution raises our concerns in Variational Autoencoders (VAEs). The sampling distribution and the true distribution output by the encoder are different. Based on the mathematical theory of VAEs (Kingma & Welling, 2014), the encoder is utilized to encode the data distribution into a Gaussian, and the decoder is utilized to reconstruct the data distribution from the Gaussian distribution. In particular, the reconstruction loss is used for the reconstruction component, and the KL divergence loss is utilized to optimize the encoder output distribution toward a Gaussian. However, in practice, the output of the encoder actually consists of features with respect to images. The addition of these features and noise constitutes a mixture of Gaussians, not a single Gaussian. In the ideal condition, when the mean values output by the encoder are pushed to zero, the mixture of Gaussians collapses to a single Gaussian. But such a condition goes against the reconstruction process, which requires more information for accurate reconstruction. It is clear that the KL loss and the reconstruction loss in VAEs are in conflict with each other. The essence of this conflict is the fit–overfit tension we mentioned above. The VAE utilizes the reconstruction process to fit the empirical distribution, while the KL term prevents the model from becoming a memorization model. Therefore, as long as the reconstruction loss exists, it is almost impossible for the latent space of a VAE to become a single Gaussian. To demonstrate this, we utilized a standard VAE implementation from GitHub, set the latent dimension to two, and used 64 images for training. In particular, we assigned different weights to the KL loss and plotted latent samples to display the true distribution used as input to the decoder. The results are shown in Fig. 6. It is clear that the true distribution in latent space is a mixture of Gaussians rather than a single Gaussian, whereas sampling uses a single Gaussian. The logical inconsistency of VAEs is thus established.

### A.1.5 ONE-TO-MANY NON-CONVERGENCE

In neural networks that involve stochasticity, a specific input to a network may correspond to multiple valid outputs. For a purely deterministic mapping, such one-to-many situations create optimization conflicts, since a function cannot simultaneously map one input to several targets. Generative models often face this condition. For example, in VAEs, noise is added to the encoder output, so two different feature vectors with their respective noise can produce the same input for the decoder, which is then optimized toward different training images. In this case, the encoder tends to minimize the expected reconstruction error, effectively learning an averaged or fused representation. Diffusion models exhibit similar behavior, and autoregressive models encounter it even more frequently, as the same token context can lead to several different next tokens. An important issue of such one-to-many mappings is that the training loss cannot be reduced to zero. This does not necessarily mean divergence, but rather reflects an irreducible loss arising from the intrinsic uncertainty of the mapping. As a result, inference may appear unstable: when the loss does not vanish, it is unclear whether the residual error is due to this inherent non-determinism or to insufficient model capacity.

### A.1.6 AUTOREGRESSIVE MODELS TO MEMORIZATION

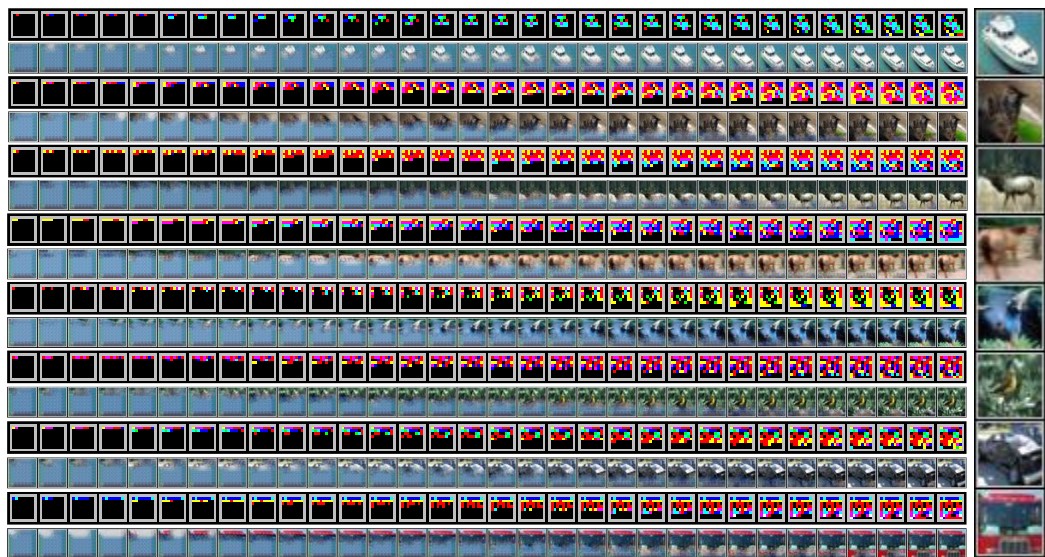

Figure 8: The inference process of PixelCNN trained with 64 images. All generated images closely resemble the training images, resulting in a UIR value of 0%.

When the training data is finite and the length of each training instance is also finite (e.g., image data), an autoregressive model tends to become a memorization model without locality, provided it has sufficient learning capacity. The key idea of an autoregressive model is to predict the next token. A very common expression of autoregressive models is (van den Oord et al., 2016):

$$p(\mathbf{x}) = \prod_{i=1}^{N} p(x_i \mid x_1, \cdots, x_{i-1}). \quad (12)$$

This is basically a factorization of the total probability. When the observation range includes all of the previous tokens, the final target becomes

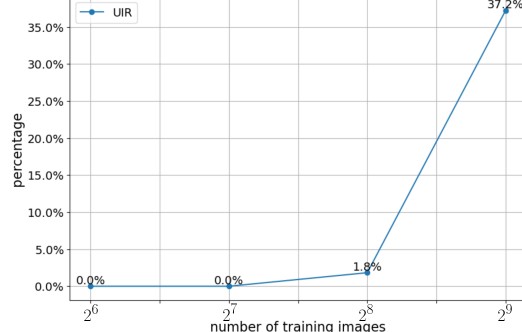

Figure 7: The UIR of PixelCNNs trained with different numbers of training images.

the empirical distribution under the condition of finite training data. To demonstrate this, we utilized the public implementation of PixelCNN (van den Oord et al., 2016), fed it with our binary latents

after a finite number of sampling trials. We then computed the UIR of the generated images of PixelCNN with a PSNR threshold of 30. During training, we used cross-entropy loss for optimization, since our tokens are discrete values. The results are shown in Fig. 7, and the qualitative results are shown in Fig. 8. As shown in Fig. 7, when only $2^6 = 64$ images are used for training, the UIR of PixelCNN is 0, which means that all generated images are training images. When the number of training images increases to $2^8$, a few generated images differ from the training images. When the number of training images increases to $2^9$, many generated images differ from the training images. However, such conditions mainly arise because the PixelCNN we utilized does not have enough parameters. If the network had sufficient parameters to provide enough learning capacity, it would become a memorization model. Another concern is the regularization of the network, but as we discussed before, such regularization is largely based on subjective biases. In addition, regularization also restricts the learning capability of networks. Therefore, locality is the key to guaranteeing the generative ability of autoregressive models.

### A.1.7 EXCLUSIVITY OF PROBABILITY SPACES

The exclusivity of the overlap between the probability spaces of random variables derives from the one-to-many non-convergence proposed by Zhao & Basu (2025). When the probability spaces of different random variables exhibit overlap, there is a possibility that the observation values within the overlapping range will be optimized with respect to the targets corresponding to both random variables. Under such conditions, one-to-many non-convergence emerges, and the network loss cannot reach zero. To address this, the network tends to reduce such situations. Consequently, the distance between the centers of the random variables increases, pushing them away from each other. In this way, the exclusivity of their probability spaces is established. To further demonstrate this, we proposed a toy experiment in Fig. 9. In particular, 1000 images from CIFAR-10 are used for training, and the latent dimension is set to 1, so the latents can be visualized as shown in the right part of Fig. 9. The plots of epochs versus average MSE, average PSNR, average latent value, and average latent binary rate are shown in the left part of Fig. 9. At epoch 8000, we applied the indicator functions, and all the values of the latent set became signed binary.

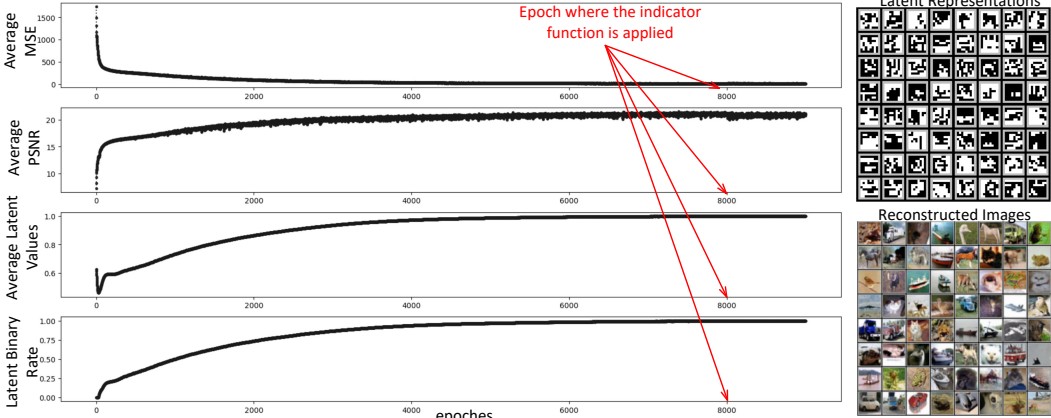

Figure 9: The demonstration of capturing signed binary latents.

### A.2 MATHEMATICAL DERIVATION

### A.2.1 DERIVATION OF EQ. 7

The random variables $\tilde{\mathbf{z}}^{\pm} = 2 \cdot \mathbf{1}[\mathbf{z} + \epsilon > 0] - 1$ in the proposed approach is basically an independent (but not identically distributed) Bernoulli/Rademacher vector, controlled by the mean value $\mathbf{z}$, whose probability mass function is:

$$P(\tilde{\mathbf{z}} = \mathbf{k}) = (1 - \pi)^{||\mathbf{z} \oplus \mathbf{k}||_1} \cdot \pi^{||\mathbf{z}||_0 - ||\mathbf{z} \oplus \mathbf{k}||_1}, \tag{13}$$

In a similar manner, the probability mass function in the first formula in Eq. 7 can be written as:

$$p(\mathbf{z}_i^\pm|\tilde{\mathbf{G}}_n^\pm) = P(\mathbf{z}_i^\pm = \mathbf{k}|\text{Mid}(\tilde{\mathbf{G}}_n^\pm) = \mathbf{c}_n) = (1-\pi)^{\|\mathbf{k}\oplus\mathbf{c}_n\|_1} \cdot \pi^{\|\mathbf{k}\|_0 - \|\mathbf{k}\oplus\mathbf{c}_n\|_1}. \tag{14}$$

The probability mass function of the second formula in Eq. 7 needs using bays rule [4]:

$$p(\tilde{\mathbf{G}}_n^\pm|\mathbf{z}_{[i-\rho,i)}^\pm) = \frac{p(\mathbf{z}_{[i-\rho,i)}^\pm|\tilde{\mathbf{G}}_n^\pm)p(\tilde{\mathbf{G}}_n^\pm)}{p(\mathbf{z}_{[i-\rho,i)}^\pm)}. \tag{15}$$

In particular, both $p(\tilde{\mathbf{G}}_n^\pm) = \frac{1}{N}$, and $p(\mathbf{z}_{[i-\rho,i)}^\pm) = P(\mathbf{z}^\circ)$ are constant values. $p(\mathbf{z}_{[i-\rho,i)}^\pm)$ can be computed with the similar manner in Eq. 13. Then we have:

$$\begin{aligned}
p(\tilde{\mathbf{G}}_n^\pm|\mathbf{z}_{[i-\rho,i)}^\pm) &= P(\text{Pre}(\tilde{\mathbf{G}}_n^\pm) = \mathbf{g}_n \mid \mathbf{z}_{[i-\rho,i)}^\pm = \mathbf{z}^\circ) \\
&= \frac{P(\mathbf{z}_{[i-\rho,i)}^\pm = \mathbf{z}^\circ|\text{Pre}(\tilde{\mathbf{G}}_n^\pm) = \mathbf{g}_n)P(\text{Pre}(\tilde{\mathbf{G}}_n^\pm) = \mathbf{g}_n)}{P(\mathbf{z}_{[i-\rho,i)}^\pm = \mathbf{z}^\circ)} \\
&= \frac{(1-\pi)^{\|\mathbf{g}_n\oplus\mathbf{z}^\circ\|_1} \cdot \pi^{\|\mathbf{z}^\circ\|_0 - \|\mathbf{g}_n\oplus\mathbf{z}^\circ\|_1}}{N \cdot P(\mathbf{z}^\circ)}
\end{aligned} \tag{16}$$

### A.2.2 DERIVATION OF EQ. 8

$\exp(\cdot)$ denotes the natural exponential function and $\log(\cdot)$ denotes the natural logarithm (base $e$), so that $\exp(\log(x)) = x$ for $x > 0$. By plugging Eq. 7 into Eq. 6, we have:

$$\begin{aligned}
&\sum_{n=1}^N p(\mathbf{z}_i^\pm|\tilde{\mathbf{G}}_n^\pm)p(\tilde{\mathbf{G}}_n^\pm|\mathbf{z}_{[i-\rho,i)}^\pm) \\
&= \sum_{n=1}^N P(\mathbf{z}_i^\pm = \mathbf{k}|\text{Mid}(\tilde{\mathbf{G}}_n^\pm) = \mathbf{c}_n)P(\text{Pre}(\tilde{\mathbf{G}}_n^\pm) = \mathbf{g}_n|\mathbf{z}_{[i-\rho,i)}^\pm = \mathbf{z}^\circ) \\
&= \sum_{n=1}^N P(\mathbf{k}|\mathbf{c}_n)P(\mathbf{g}_n|\mathbf{z}^\circ) = \sum_{n=1}^N \exp\Big(\log\big(P(\mathbf{k}|\mathbf{c}_n)P(\mathbf{g}_n|\mathbf{z}^\circ)\big)\Big) \\
&= \sum_{n=1}^N \exp\Big(\log\big(P(\mathbf{k}|\mathbf{c}_n)\big) + \log\big(P(\mathbf{g}_n|\mathbf{z}^\circ)\big)\Big) = \sum_{n=1}^N \exp\Big(\log\big(P(\mathbf{k}|\mathbf{c}_n)\big) + \log\big(\frac{P(\mathbf{z}^\circ|\mathbf{g}_n)P(\mathbf{g}_n)}{P(\mathbf{z}^\circ)}\big)\Big) \\
&= \sum_{n=1}^N \exp\Big(\log\big(P(\mathbf{k}|\mathbf{c}_n)\big) + \log\big(P(\mathbf{z}^\circ|\mathbf{g}_n)\big) + \log\big(\frac{P(\mathbf{g}_n)}{P(\mathbf{z}^\circ)}\big)\Big) \\
&= \sum_{n=1}^N \exp\Big(\log\big(P(\mathbf{k}|\mathbf{c}_n)\big) + \log\big(P(\mathbf{z}^\circ|\mathbf{g}_n)\big)\Big) \cdot \exp\Big(\log\big(\frac{P(\mathbf{g}_n)}{P(\mathbf{z}^\circ)}\big)\Big) \quad \because \frac{P(\mathbf{g}_n)}{P(\mathbf{z}^\circ)} = \frac{1}{NP(\mathbf{z}^\circ)} \\
&= \frac{1}{NP(\mathbf{z}^\circ)} \cdot \sum_{n=1}^N \exp\Big(\log\big(P(\mathbf{k}|\mathbf{c}_n)\big) + \log\big(P(\mathbf{z}^\circ|\mathbf{g}_n)\big)\Big) \\
&= \frac{1}{NP(\mathbf{z}^\circ)} \cdot \sum_{n=1}^N \exp\Big(\log\big((1-\pi)^{\|\mathbf{k}\oplus\mathbf{c}_n\|_1} \cdot \pi^{\|\mathbf{k}\|_0 - \|\mathbf{k}\oplus\mathbf{c}_n\|_1}\big) + \log\big((1-\pi)^{\|\mathbf{g}_n\oplus\mathbf{z}^\circ\|_1} \cdot \pi^{\|\mathbf{z}^\circ\|_0 - \|\mathbf{g}_n\oplus\mathbf{z}^\circ\|_1}\big)\Big) \\
&= \frac{1}{NP(\mathbf{z}^\circ)} \cdot \sum_{n=1}^N \exp\Big(\big(\|\mathbf{k}\oplus\mathbf{c}_n\|_1 + \|\mathbf{g}_n\oplus\mathbf{z}^\circ\|_1\big)\log(1-\pi) + \big(\|\mathbf{k}\|_0 - \|\mathbf{k}\oplus\mathbf{c}_n\|_1 + \|\mathbf{z}^\circ\|_0 - \|\mathbf{g}_n\oplus\mathbf{z}^\circ\|_1\big)\log\pi\Big) \\
&\qquad \text{Let} \quad \lambda = \|\mathbf{k}\|_0 + \|\mathbf{z}^\circ\|_0, \qquad \eta(\mathbf{k};\tilde{\mathbf{G}}_n^\pm,\mathbf{z}^\circ) = \|\mathbf{k}\oplus\mathbf{c}_n\|_1 + \|\mathbf{g}_n\oplus\mathbf{z}^\circ\|_1 \\
&= \frac{1}{NP(\mathbf{z}^\circ)} \cdot \sum_{n=1}^N \exp\Big(\eta(\mathbf{k};\tilde{\mathbf{G}}_n^\pm,\mathbf{z}^\circ)\log(1-\pi) + \big(\lambda - \eta(\mathbf{k};\tilde{\mathbf{G}}_n^\pm,\mathbf{z}^\circ)\big)\log\pi\Big)
\end{aligned} \tag{17}$$

---

[4]Bayes' rule is a mathematical identity; employing it does not conflict with our Frequentist perspective.

Next, a normalization with respect to the final label $\mathbf{k}$ is applied to make the computation stable. The final expression of Eq. 8 is captured: Mathematically:

$$P(\mathbf{k}|\mathbf{z}^\circ) = \frac{P(\mathbf{k}|\mathbf{z}^\circ)}{\sum_{\mathbf{k}'} P(\mathbf{k}'|\mathbf{z}^\circ)} = \frac{\sum_{n=1}^{N} \exp\left(\eta(\mathbf{k}; \tilde{\mathbf{G}}_n^\pm, \mathbf{z}^\circ)\log(1-\pi) + \left(\lambda - \eta(\mathbf{k}; \tilde{\mathbf{G}}_n^\pm, \mathbf{z}^\circ)\log\pi\right)\right)}{\sum_{\mathbf{k}'}\sum_{n=1}^{N} \exp\left(\eta(\mathbf{k}'; \tilde{\mathbf{G}}_n^\pm, \mathbf{z}^\circ)\log(1-\pi) + \left(\lambda - \eta(\mathbf{k}'; \tilde{\mathbf{G}}_n^\pm, \mathbf{z}^\circ)\log\pi\right)\right)} \quad (18)$$

Noted that in order to further make the computation stable, we utllized the log-sum-exp trick, which is a common numerical issue arises when evaluating expressions of the form $\log\left(\sum_{n=1}^{N}\exp(x_n)\right)$. Because we want to keep all probability into log form, and only apply $\exp(\cdot)$ on the last step before sampling. Mathematically:

$$\log\left(\sum_{n=1}^{N}\exp(x_n)\right) = M + \log\left(\sum_{n=1}^{N}\exp(x_n - M)\right), \quad (19)$$

$$P(\mathbf{k} \mid \mathbf{z}^\circ) = \frac{\sum_{n=1}^{N} \exp\left(\psi(\mathbf{k}, \tilde{\mathbf{G}}_n, \mathbf{z}^\circ)\right)}{\sum_{\mathbf{k}'}\sum_{n=1}^{N} \exp\left(\psi(\mathbf{k}', \tilde{\mathbf{G}}_n, \mathbf{z}^\circ)\right)} \quad (20)$$

where $M = \max_n(x_n)$, $\psi(\mathbf{k}, \tilde{\mathbf{G}}_n, \mathbf{z}^\circ) = \eta(\mathbf{k}; \tilde{\mathbf{G}}_n^\pm, \mathbf{z}^\circ)\log(1-\pi) + \left(\lambda - \eta(\mathbf{k}; \tilde{\mathbf{G}}_n^\pm, \mathbf{z}^\circ)\log\pi\right)$ The ultimate expression used to compute the probability for sampling is:

$$P(\mathbf{k} \mid \mathbf{z}^\circ) = \exp\left(\log\left(P(\mathbf{k}|\mathbf{z}^\circ)\right)\right) = \exp\left(\Psi(\mathbf{k}) + \log\left(\sum_{n=1}^{N}\exp\left(\psi(\mathbf{k}, \tilde{\mathbf{G}}_n, \mathbf{z}^\circ) - \Psi(\mathbf{k})\right)\right)\right.$$
$$\left. - \sum_{\mathbf{k}'}\left[\Psi(\mathbf{k}') + \log\left(\sum_{n=1}^{N}\exp\left(\psi(\mathbf{k}', \tilde{\mathbf{G}}_n, \mathbf{z}^\circ) - \Psi(\mathbf{k}')\right)\right)\right]\right). \quad (21)$$

where $\Psi(k) = \max_n \psi(\mathbf{k}, \tilde{\mathbf{G}}_n, \mathbf{z}^\circ)$. The computation stability is very important to the proposed approach. Especially when the direct sampling is used for image generation. It is main reasons we utilized the log-sum-exp trick.

### A.2.3 TRIANGULAR DISTRIBUTION TO RADEMACHER DISTRIBUTION

In work, we utilize the indicator function to transform random varialbes $\tilde{\mathbf{z}}$ following Triangular distribution to $\tilde{\mathbf{z}}^\pm$ following Rademacher distribution. Before transformaion, we have $\tilde{\mathbf{z}} = \mathbf{z} + \tau \cdot \epsilon$. In particular, the probability density function of $\epsilon$ is:

$$\epsilon \sim \text{Tri}(\kappa) = \begin{cases} (1 - u^\kappa), & \text{if } u > 0^+ \\ (|u|^\kappa - 1), & \text{if } u < 0^-, \end{cases} \quad (22)$$

where $u \sim \mathcal{U}(-1, 1)$ denotes the uniform distribution over $[-1, 1]$. Since the values of $\mathbf{z}$ are -1 or 1. We discuss the conditions when $\mathbf{z} = -1$. Consider the affine transform:

$$\tilde{\mathbf{z}} = -1 + \tau \cdot \epsilon. \quad (23)$$

When $\tau \leq 1$, the value of $\tilde{\mathbf{z}}^\pm$ always smaller or equal to $0^-$. Therefore,

$$\pi = \mathbb{P}(\tilde{\mathbf{z}}^\pm \leq 0) = 1 \iff \mathbb{P}\left((\mathbf{1}[\tilde{\mathbf{z}} > 0] - 1) = -1\right) = 1. \quad (24)$$

Random varialbes $\tilde{\mathbf{z}}^\pm$ become deterministic variables. When $\tau > 1$. Since the distribution is continuous and the singleton $\{0\}$ is a null set (zero Lebesgue measure), the probability mass is entirely concentrated on the negative and positive half-lines. Hence $\mathbb{P}(\tilde{\mathbf{z}}^\pm = 0) = 0$, $\mathbb{P}((-\infty, 0)) = \mathbb{P}(\tilde{\mathbf{z}}^\pm < 0)$ and $\mathbb{P}((0, \infty)) = \mathbb{P}(\tilde{\mathbf{z}}^\pm > 0)$. We have:

$$\tilde{\mathbf{z}}^\pm > 0 \iff \epsilon > \frac{1}{\tau}. \quad (25)$$

For any $t \in (0,1)$, the (right) tail of $\varepsilon$ satisfies

$$\mathbb{P}(\epsilon > t) = \mathbb{P}\big(u > 0,\ 1 - u^\kappa > t\big) = \mathbb{P}\big(0 < u < (1-t)^{1/\kappa}\big) = \frac{1}{2}\,(1-t)^{1/\kappa}.$$

Substituting $t = \frac{1}{\tau}$ we have:

$$\pi = \mathbb{P}\big(\tilde{\mathbf{z}}^\pm \le 0\big) = 1 - \mathbb{P}\big(\tilde{\mathbf{z}}^\pm > 0\big) = 1 - \frac{1}{2}\left(1 - \frac{1}{\tau}\right)^{1/\kappa}. \tag{26}$$

With the similr manner when $\mathbf{z} = 1$, we have the same expression of $\pi$.

### A.3 DEMONSTRATION OF PROPERTIES

#### A.3.1 FIDELITY AND PATCH CONSISTENCY

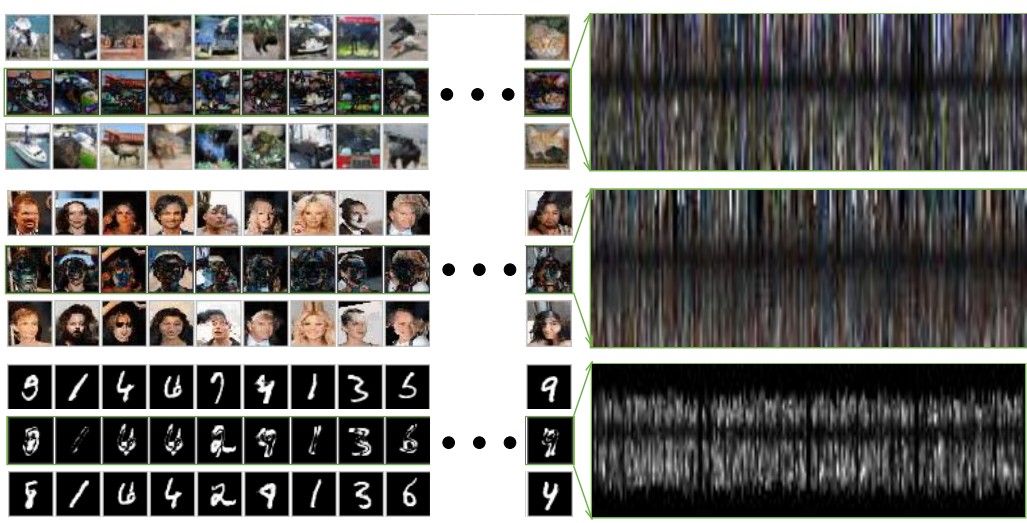

Figure 10: Demonstration of patch consistency in CIFAR-10, CelebA and MNIST datasets.

To demonstrate the patch consistency, we train the model on different datasets to obtain latent representations for each image. In particular, the size of latents is $7{\times}8{\times}8$. For a pair of images $\mathbf{z}^-$ and $\mathbf{z}^+$, we construct new latents by concatenating the upper part ($\mathbf{z}^-[:,1{:}3,:]$), the center row ($\mathbf{z}^-[:,4,:]$), and the lower part ($\mathbf{z}^+[:,5{:}7,:]$) of their latent representations, and decode the new latents via $d_\theta(\cdot)$ to generate a new reconstruction $\mathbf{X}_+^-$. Conversely, we swap the source of the upper/lower parts to obtain $\mathbf{X}_-^+$, which is:

$$\begin{aligned}
\mathbf{X}_+^- &= d_\theta([\mathbf{z}^-[:,1{:}3,:];\mathbf{z}^-[:,4,:];\mathbf{z}^+[:,5{:}7,:]]) \\
\mathbf{X}_-^+ &= d_\theta([\mathbf{z}^+[:,1{:}3,:];\mathbf{z}^+[:,4,:];\mathbf{z}^-[:,5{:}7,:]]).
\end{aligned} \tag{27}$$

Next, we compute the difference map as $\mathbf{D} = |\mathbf{X}_+^- - \mathbf{X}_-^+|$. For multiple pairs, we sum each difference map along columns and concatenate all results to form the final visualization $\mathbf{M}$, which is:

$$\mathbf{M} = \operatorname*{concat}_{i} \sum_{\text{col}} \mathbf{D}^{(i)}, \tag{28}$$

where $\operatorname{concat}_i$ denotes concatenation over samples, and $\sum_{\text{col}}$ denotes summation over columns. As shown in Fig. 10, a prominent dark line appears at the center of the difference maps, which demonstrates the patch consistency. The generated images in the proposed approach are thus a reorganization of the existing features or patches in the training images. In particular, the consistency is ensured by the reconstruction quality. Therefore, it is guaranteed that the patches in the generated images come from the training images, and the fidelity of the proposed approach is thus exhibited.

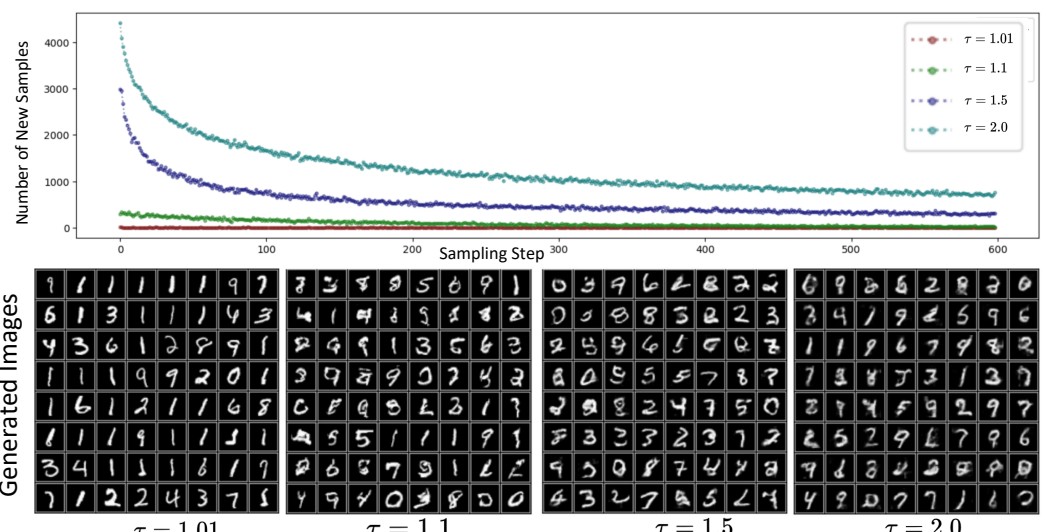

Figure 11: Demonstration of controllable diversity in generation based on exhaustive search.

### A.3.2 CONTROLLABLE DIVERSITY

To demonstrate controllable diversity, we varied the values of $\tau$ to adjust the probability $\pi$, while keeping $\kappa$ fixed at 2. In this setting, the uncertainty of the random variables increases with the increasing of $\tau$. It means that more new samples are generated in each sampling trial, as shown in Fig. 11. We performed 600 sampling runs to create a sufficient number of token pairs for generation in which exhaustive search approach is used. When $\tau = 1.01$, the uncertainty is very low, resulting in a limited number of new samples per step. In this case, the generated diversity is poor, but the fidelity is very high, which is preserved by the reconstruction quality. As $\tau$ increases, more new samples are produced, but the decoder's reconstruction quality degrades compared to the cases with smaller $\tau$ values. Consequently, diversity increases while fidelity decreases. Note that, besides the value of $\tau$, the observation range $\rho$ also influences the generation diversity. In the extreme case, when the context length is sufficiently large to model the global distribution, an autoregressive model tends to memorize the training set.

### A.3.3 NETWORK APPROXIMATION VS DIRECT SAMPLING

In this paper, most of the generated images are obtained through direct sampling. However, it is also possible to utilize a network to approximate the sampling distribution. Unfortunately, training a network to approximate the distribution of locally dependent random variables usually requires sufficient learning capacity, which results in a high computational cost. Therefore, the images generated by the network we trained are not as good as those generated by direct sampling. To demonstrate this, we trained a network for approximation. The architecture we utilized is a simple multilayer perceptron with cross-entropy loss. In particular, 1000 images from MNIST were used for training. The results are shown in Fig. 12. When the observation range is 1, the results of the network contain barely any training images. This actually demonstrates that the network is not well trained. Compared with network approximation, direct sampling already includes a few training images. When the observation range increases to 3 and 5, some training images appear. But even when the observation range is 7, which corresponds to the global distribution, the network is still not perfect. This is due to the one-to-many non-convergence. Meanwhile, direct sampling consists almost entirely of training images, with barely any errors. It should be noted that the observation range also affects the reconstruction quality, which is the main reason small errors occur in the images with an observation range of 5. In fact, considering the flexibility of network approximation, training a network could be a better approach than direct sampling, under the condition of sufficient computational resources.

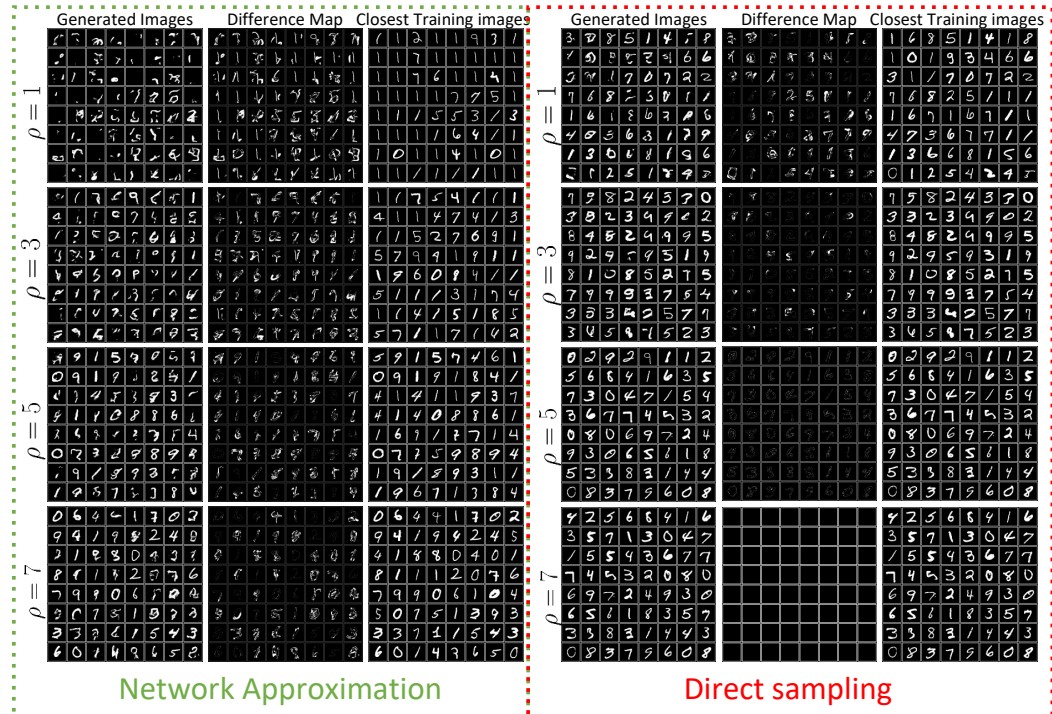

Figure 12: Comparison between the generated images from network approximation and direct sampling, with different observation ranges.

### A.3.4 IMAGE GENERATION WITH 64 TRAINING IMAGES

The proposed approach is able to perform image generation with a limited number of images. We used 64 images for generation. We first randomly selected 64 images from the training set. We then fed these images into the autoencoder to create locally dependent random variables, which were subsequently used for direct sampling. We evaluated generation with a limited number of images on MNIST, CIFAR-10, and CelebA. In particular, the MNIST and CIFAR-10 images were resized to $32 \times 32$, and the CelebA images were resized to $64 \times 64$. The generation results are shown in Fig. 13. The generation quality on MNIST is quite good. The main reason is that the images in MNIST share many similar local patterns, so the latents are closer to each other. The generation on CelebA is also acceptable, but not as good as on MNIST, since color images contain more information. The generation quality on CIFAR-10 is relatively limited, as the 64 training images do not share many similar patterns. However, the latent variables still make the patches close to each other, which contributes to the generation quality. To quantitatively demonstrate the generation quality, we evaluated the FID between the unique generated images (with a PSNR filter of 30) and the entire CIFAR-10 dataset. The FID score is 138, which is quite promising. Note that this FID score is not entirely fair, because the global distribution has changed. To demonstrate this, we also computed the FID between these 64 training images and the entire CIFAR-10 dataset. The FID score is 175, which is even higher than that based on the unique generated images. When we compute the FID between the generated images and their closest training images, the FID decreases to 70. This phenomenon demonstrates that FID is significantly influenced by similarity to empirical distributions, which further highlights the fairness issue of using FID for comparing generative models.

### A.3.5 HIGH-RESOLUTION IMAGE GENERATION

The proposed approach is able to generate high-resolution images. Unfortunately, due to the limitation of our computational resources, we were unable to evaluate the proposed approach on the entire dataset. Instead, 200 images of size $218 \times 178$ from the CelebA dataset were used, with a latent size of $13 \times 11$. The qualitative results are shown in Fig. 14. We computed the FID between the generated images and the training images, using a PSNR filter with a threshold of 25. The FID is 84.

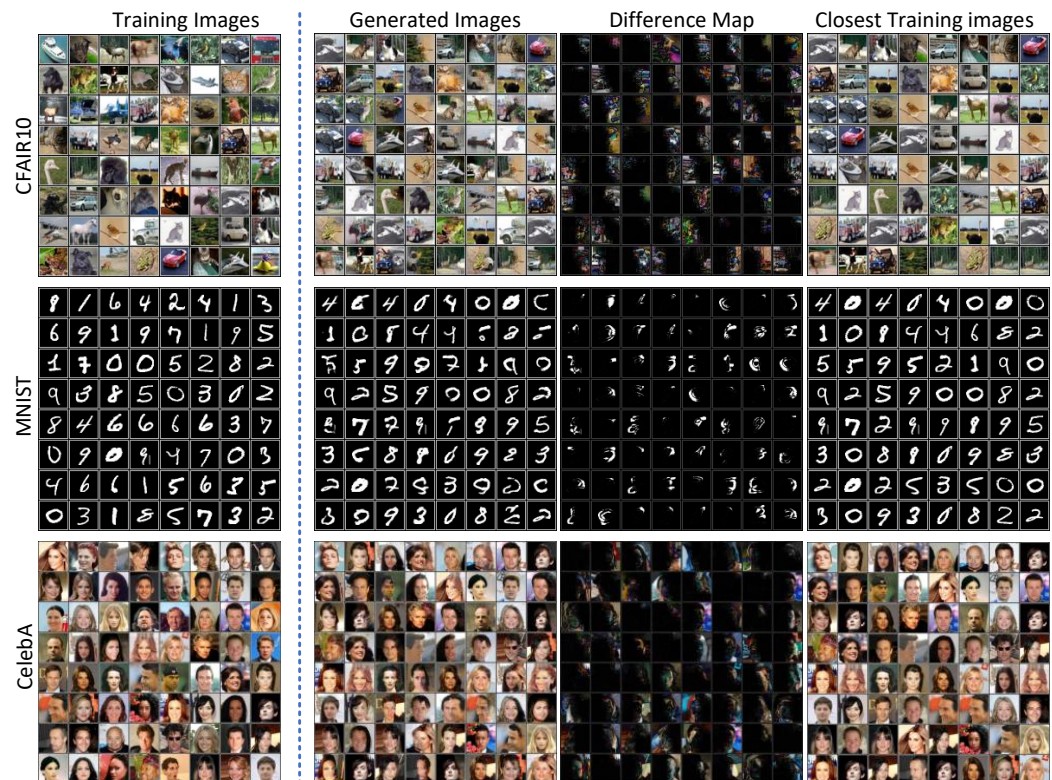

Figure 13: Demonstration of generated images using only 64 training images on the CIFAR-10, MNIST, and CelebA datasets. In particular, the FID between the unique generated images and the entire CIFAR-10 dataset is 138.

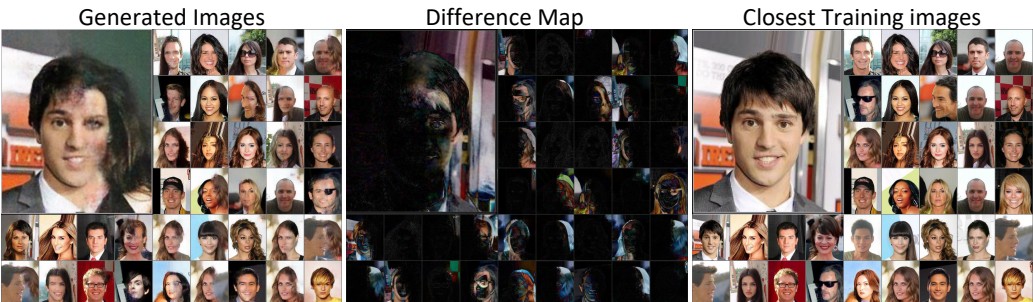

Figure 14: Demonstration of generated images with a high resolution of $218 \times 178$, using direct sampling for generation. The FID between the generated images and the training images is 84.

The reason for this relatively poor FID is the limitation of our computational resources. The latent size was restricted to $13 \times 11$, and only 200 images were used for training, which is far fewer than the total number of images in CelebA. With a larger latent size, the reconstruction quality would be better, which would in turn improve the generation quality. For example, as shown in Fig. 14, the stitching boundaries are not smooth enough; this is because the reconstruction quality of the autoencoder is not sufficient.

### A.3.6 CERTIFIABILITY

To demonstrate the certifiability of the proposed approach, we present all possible images generated under the strategies of exhaustive search and direct sampling. First, we used 64 images from MNIST for training to generate locally dependent random variables, with $1 \times 8 \times 8$ as the size of the latent

variables. Then, we performed sampling with 1 to 10 trials to obtain samples from the random variables, which were subsequently fed into the decoder for further training. The number of samples increased with more sampling trials, and after further training of the decoder, the empirical distribution replaced the true distribution of locally dependent random variables. These empirical distributions with different numbers of samples were then used for generation. With more sampling trials, the number of samples in the empirical distribution increases, which in turn increases the number of possible combinations of latents, resulting in more generated images and more unique generated images. The plot of the number of generated images and the number of unique generated images is shown in Fig. 15. When only one sampling trial is used, the number of generated images is limited to 69. Note that 64 of these images are training images, and only 5 are newly generated. These 5 new images are very close to certain training images. When the number of sampling trials increases to 10, there are over 400 generated images. All of them can be displayed for certification, as shown in Fig. 16. By applying a PSNR filter with a value of 30, the new images are highlighted in Fig. 17. All of the generated images can thus be certified, and all newly generated images are explicitly displayed. Therefore, the certifiability of the proposed approach with exhaustive search is demonstrated.

To further demonstrate certifiability in a general manner, we re-trained the network with a smaller latent size of $1 \times 3 \times 3$. In this condition, since the latent has a size of $1 \times 3 \times 3$ and each entry can take only two possible values ($-1$ or $1$), the total number of possible latent configurations is $2^{3 \times 3} = 512$. We exhaustively enumerated all $2^{3 \times 3}$ possible assignments of the latent and generated the corresponding images, so that every possible generated image was obtained and displayed in Fig. 18. The quality of some of the generated images is fairly good, while others are not. The appearance of these generated images is restricted by their probabilities. The generated images with good quality have higher probabilities, since their corresponding latents have a higher probability of being optimized with the training images. This makes them closer to the training images, thereby achieving better fidelity. In contrast, images with poor quality have

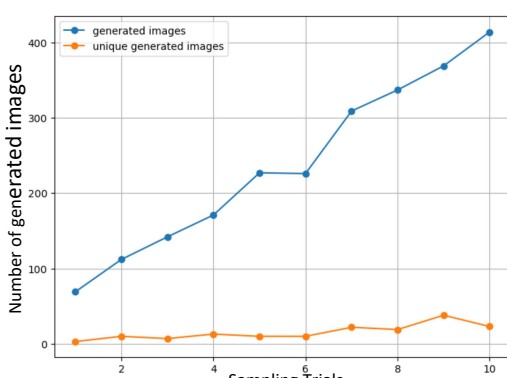

Figure 15: By repeating the sampling trials, more samples are captured. Consequently, both the number of generated images and the number of unique generated images increase.

lower probabilities, but their diversity is better. To demonstrate this, we repeatedly performed sampling trials to check which generated images appeared. The results are shown in Fig. 19. When 640 sampling trials were applied, meaning we generated 640 images in total, many of these generated images were duplicates due to the finite support of our random variables. The quality of these generated images was quite good. The unique generation results are shown in the first row of Fig. 19. As the number of sampling trials increased, more and more unique generated images appeared. When we increased the sampling trials to 64,000, which is 1000 times the number of training images, almost all of the generated images appeared, as shown in the last row of Fig. 19. This demonstrates the certifiability of the proposed approach, since all possible generated images can be exhaustively searched. The generated images are finite, and both good-quality and poor-quality images are constrained by their probabilities.

### A.3.7 QUALITATIVE GENERATION RESULTS

We qualitatively evaluated the proposed approach on CIFAR-10, MNIST, and CelebA, as shown in Fig. 20, Fig. 21, and Fig. 22, respectively.

### A.4 EXPERIMENTAL DETAILS AND NETWORK ARCHITECTURE

Network architecture is not the focus of the proposed approach; therefore, the architecture we use is ordinary and simple. Our limited computational resources are also one of the important reasons. With a better network architecture, the generation quality of the proposed approach is expected to

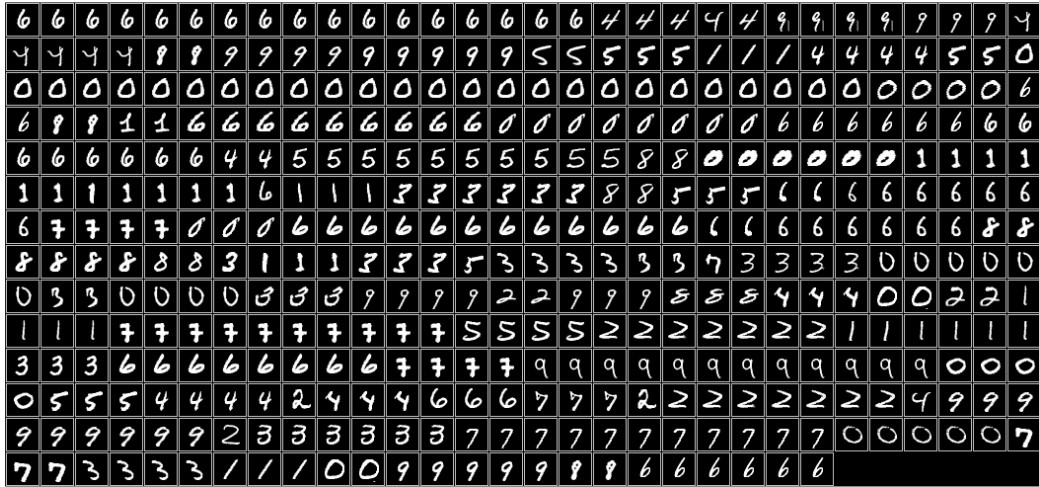

Figure 16: All possible generated images are exhaustively enumerated and displayed to demonstrate the certifiability of the proposed approach. Specifically, 64 MNIST images are used for training, and sampling is performed with 10 trials.

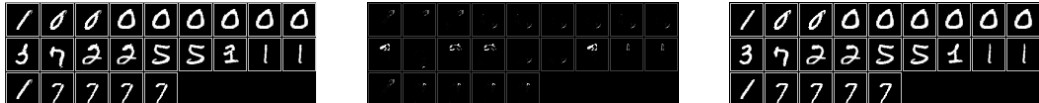

Figure 17: All unique generated images obtained with a PSNR threshold of 30, based on the results in Fig. 16.

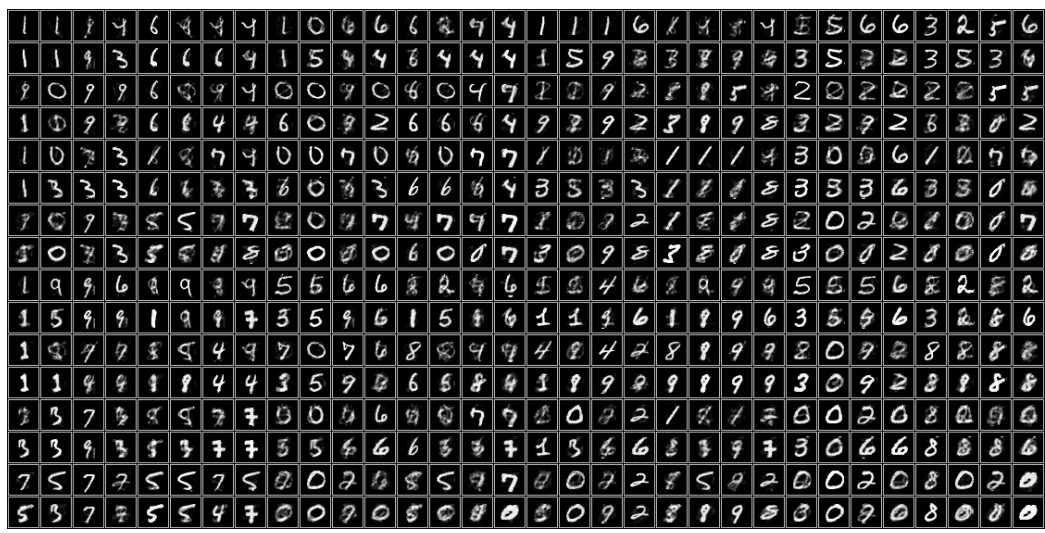

Figure 18: All possible generated images of the proposed approach with a latent size of $1 \times 3 \times 3$. The total number of possible latent values is $2^{1 \times 3 \times 3} = 512$.

improve. All experiments were run on one RTX 4090 GPU with 24GB of memory. We utilized PyTorch to implement our methods, with the Adam optimizer (Kingma & Ba, 2015) used for optimization. We employed four different architectures to create latents with sizes of $8 \times 8$, $16 \times 16$, and $32 \times 32$, as shown in Tab. 1, Tab. 2, Tab. 3, and Tab. 4. In particular, an architecture with latents of $8 \times 8$ and an observation range of $\rho = 1$ was specifically designed to demonstrate patch consistency, as shown in Tab. 1.

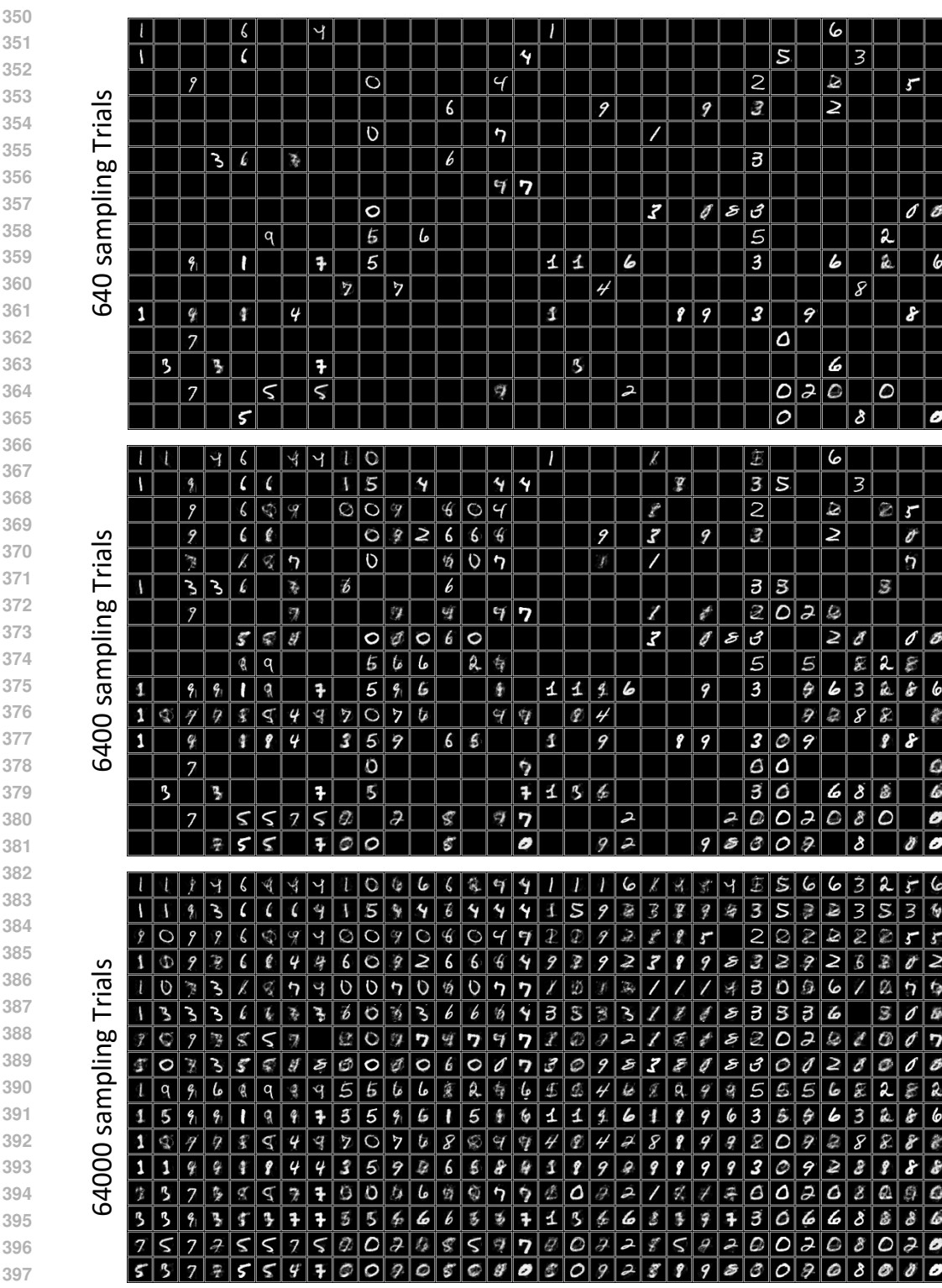

Figure 19: The appearance of generated images with different probabilities, through repeated sampling trials of 640, 6,400, and 64,000.

Generated Images    Difference Map    Closest Training images

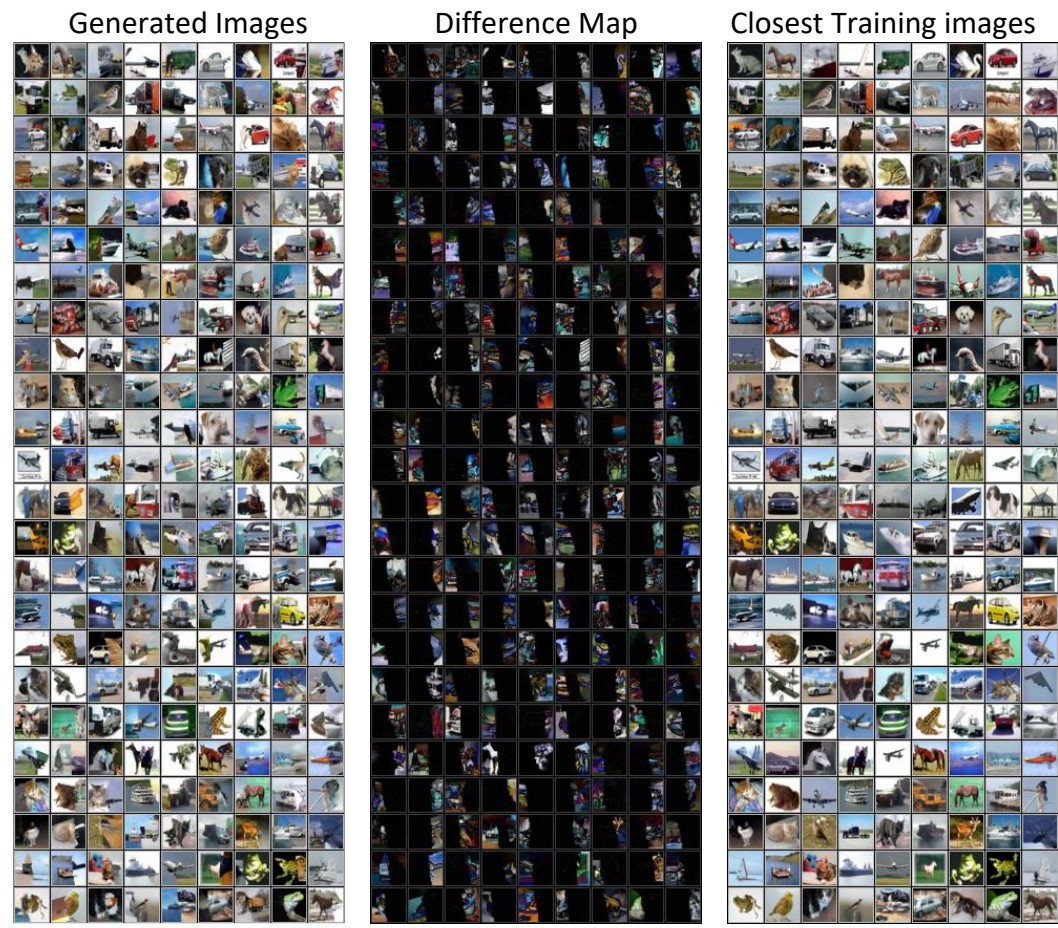

Figure 20: Images generated by the proposed approach trained on the CIFAR-10 dataset, with an FID score of 32 and a PSNR filtering threshold of 25.

## A.5 THE USE OF LARGE LANGUAGE MODELS (LLMs)

We utilized Grammarly and ChatGPT solely to check typos and grammar in the proposed paper. No technical content, experiments, or analysis were generated by large language models.

Generated Images          Difference Map          Closest Training images

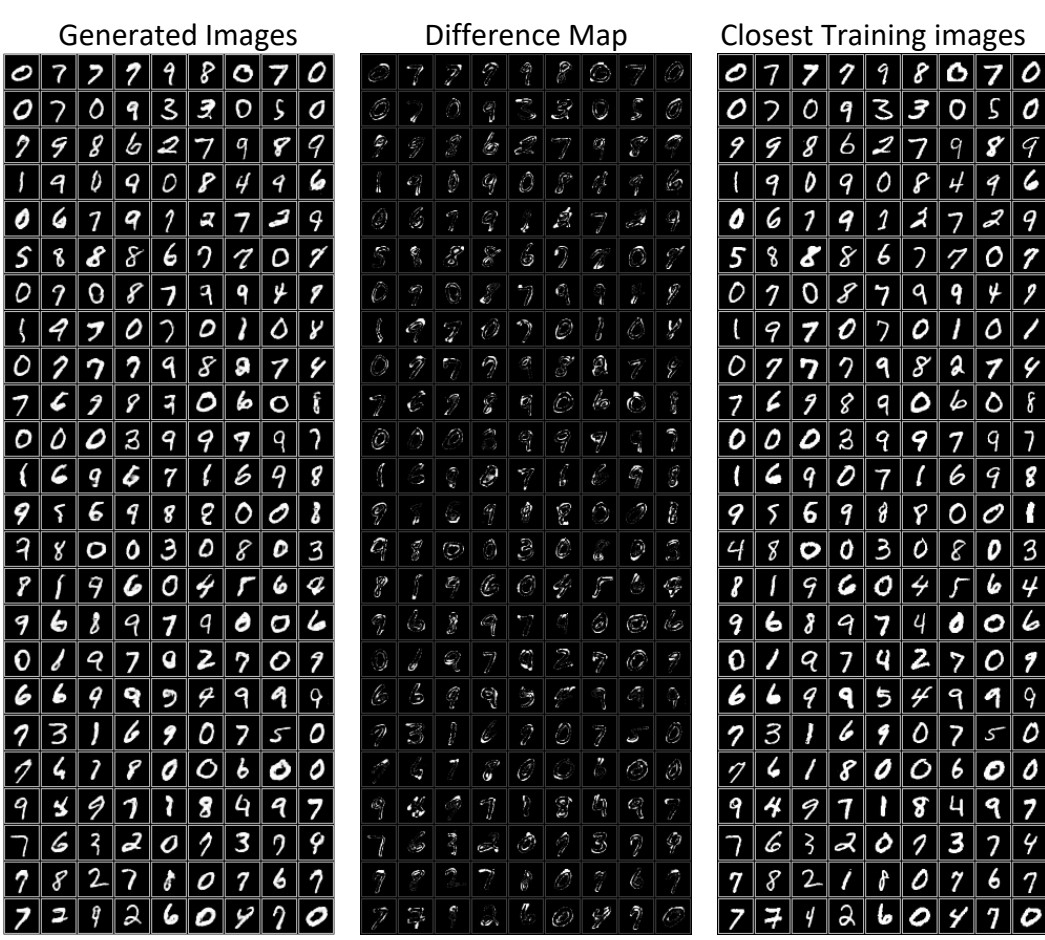

Figure 21: Images generated by the proposed approach trained on the MNIST dataset, with an FID score of 8 and a PSNR filtering threshold of 25.

Generated Images      Difference Map      Closest Training images

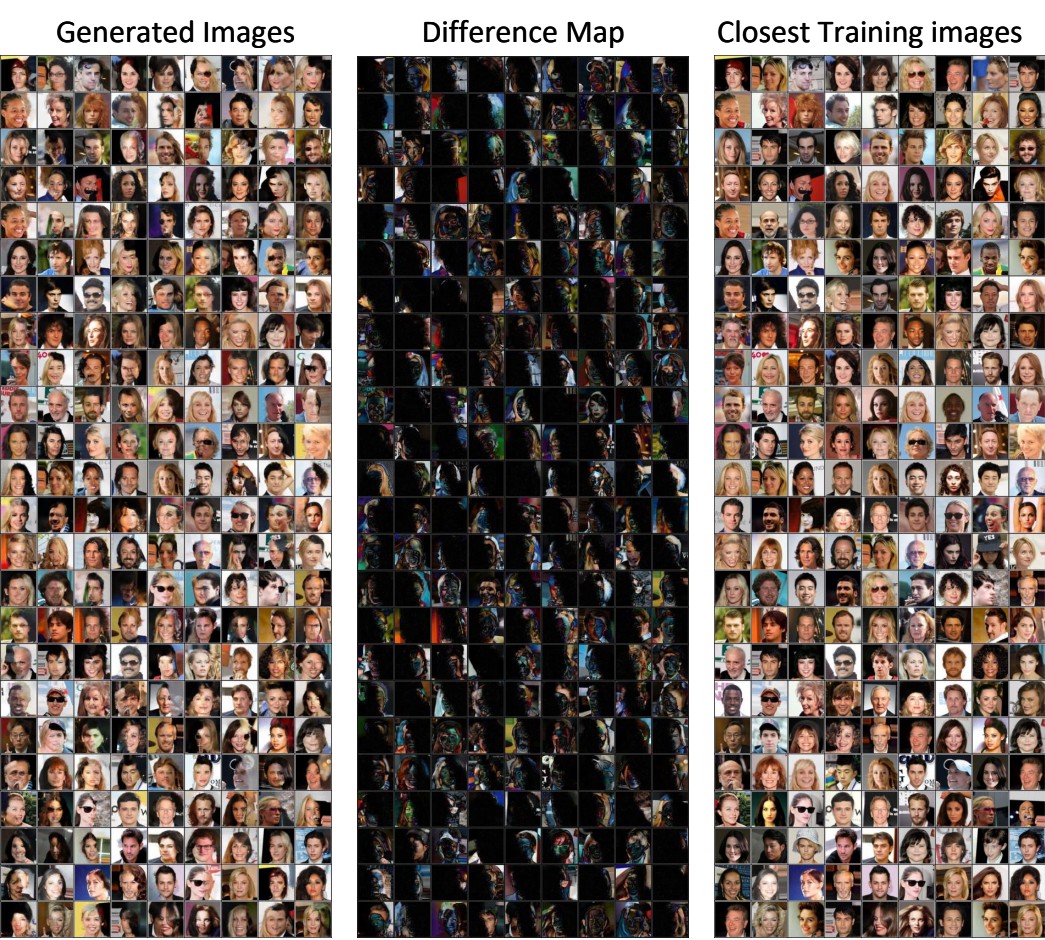

Figure 22: Images generated by the proposed approach trained on the CelebA dataset, with an FID score of 43 and a PSNR filtering threshold of 25.

Table 1: Architecture details of the autoencoder used in the proposed approach, with $\rho = 1$ and a latent size of $8 \times 8$.

| | | Type | weight | stride | padding | Data size |
|---|---|---|---|---|---|---|
| | | Input | | | | $N \times 3 \times 32 \times 32$ |
| | | Conv2d | $3 \times 32 \times 4 \times 4$ | 2 | 1 | $N \times 32 \times 16 \times 16$ |
| | | LeakyReLU | $\alpha = 0.01$ | | | $N \times 32 \times 16 \times 16$ |
| | Encoder | Conv2d | $32 \times 64 \times 4 \times 4$ | 2 | 1 | $N \times 64 \times 8 \times 8$ |
| | | LeakyReLU | $\alpha = 0.01$ | | | $N \times 64 \times 8 \times 8$ |
| | | Conv2d | $64 \times M \times 1 \times 1$ | 1 | 0 | $N \times M \times 8 \times 8$ |
| | | ResConv2d | $M \times M \times 1 \times 1$ | 1 | 0 | $N \times M \times 8 \times 8$ |
| | | ResConv2d | $M \times M \times 1 \times 1$ | 1 | 0 | $N \times M \times 8 \times 8$ |
| | Latents | | | | | $N \times M \times 8 \times 8$ |
| | | ResConv2d | $M \times M \times 1 \times 1$ | 1 | 0 | $N \times M \times 8 \times 8$ |
| | | ResConv2d | $M \times M \times 1 \times 1$ | 1 | 0 | $N \times M \times 8 \times 8$ |
| | | ConvT2d | $M \times M \times 2 \times 2$ | 2 | 0 | $N \times M \times 8 \times 8$ |
| | | Unfold | $2 \times 2$ | 1 | 1 | $N \times 4M \times 9 \times 9$ |
| | Decoder | Conv2d | $4M \times 1024 \times 1 \times 1$ | 1 | 0 | $N \times 1024 \times 9 \times 9$ |
| | | LeakyReLU | $\alpha = 0.01$ | | | $N \times 1024 \times 9 \times 9$ |
| | | Conv2d | $1024 \times 1024 \times 1 \times 1$ | 2 | 0 | $N \times 1024 \times 9 \times 9$ |
| | | LeakyReLU | $\alpha = 0.01$ | | | $N \times 1024 \times 9 \times 9$ |
| | | ConvT2d | $1024 \times 3 \times 4 \times 4$ | 4 | 2 | $N \times 3 \times 32 \times 32$ |
| | | Conv2d | $3 \times 32 \times 1 \times 1$ | 1 | 1 | $N \times 32 \times 32 \times 32$ |
| | Refine | LeakyReLU | $\alpha = 0.01$ | | | $N \times 32 \times 32 \times 32$ |
| | | Conv2d | $32 \times 3 \times 1 \times 1$ | 1 | 1 | $N \times 3 \times 32 \times 32$ |
| | | Output | | | | $N \times 3 \times 32 \times 32$ |

- **ResConv2d**: A Conv2d layer with a residual skip, i.e., $y = \text{Conv2d}(x) + x$.
- **Unfold**: PyTorch's `torch.nn.Unfold` (im2col), which extracts sliding local blocks from the input.

Table 2: Architecture details of the autoencoder used in the proposed approach, with $\rho = \lfloor \frac{R}{2} \rfloor$ and a latent size of $8 \times 8$.

| | | Type | weight | stride | padding | Data size |
|---|---|---|---|---|---|---|
| | | Input | | | | $N \times 3 \times 32 \times 32$ |
| | | Conv2d | $3 \times 32 \times 4 \times 4$ | 2 | 1 | $N \times 32 \times 16 \times 16$ |
| | | LeakyReLU | $\alpha = 0.01$ | | | $N \times 32 \times 16 \times 16$ |
| | Encoder | Conv2d | $32 \times 64 \times 4 \times 4$ | 2 | 1 | $N \times 64 \times 8 \times 8$ |
| | | LeakyReLU | $\alpha = 0.01$ | | | $N \times 64 \times 8 \times 8$ |
| | | Conv2d | $64 \times M \times 1 \times 1$ | 1 | 0 | $N \times M \times 8 \times 8$ |
| | | ResConv2d | $M \times M \times 1 \times 1$ | 1 | 0 | $N \times M \times 8 \times 8$ |
| | | ResConv2d | $M \times M \times 1 \times 1$ | 1 | 0 | $N \times M \times 8 \times 8$ |
| | Latents | | | | | $N \times M \times 8 \times 8$ |
| | | ResConv2d | $M \times M \times 1 \times 1$ | 1 | 0 | $N \times M \times 8 \times 8$ |
| | | ResConv2d | $M \times M \times 1 \times 1$ | 1 | 0 | $N \times M \times 8 \times 8$ |
| | | ConvT2d | $M \times M \times 2 \times 2$ | 2 | 0 | $N \times M \times 8 \times 8$ |
| | | Unfold | $R \times R$ | 1 | $\lfloor \frac{R}{2} \rfloor$ | $N \times M \cdot R^2 \times 8 \times 8$ |
| | Decoder | Conv2d | $M \cdot R^2 \times 1024 \times 1 \times 1$ | 1 | 0 | $N \times 1024 \times 8 \times 8$ |
| | | LeakyReLU | $\alpha = 0.01$ | | | $N \times 1024 \times 8 \times 8$ |
| | | Conv2d | $1024 \times 1024 \times 1 \times 1$ | 2 | 0 | $N \times 1024 \times 8 \times 8$ |
| | | LeakyReLU | $\alpha = 0.01$ | | | $N \times 1024 \times 8 \times 8$ |
| | | ConvT2d | $1024 \times 3 \times 4 \times 4$ | 4 | 0 | $N \times 3 \times 32 \times 32$ |
| | | Conv2d | $3 \times 32 \times 1 \times 1$ | 1 | 1 | $N \times 32 \times 32 \times 32$ |
| | Refine | LeakyReLU | $\alpha = 0.01$ | | | $N \times 32 \times 32 \times 32$ |
| | | Conv2d | $32 \times 3 \times 1 \times 1$ | 1 | 1 | $N \times 3 \times 32 \times 32$ |
| | | Output | | | | $N \times 3 \times 32 \times 32$ |

- **ResConv2d**: A Conv2d layer with a residual skip, i.e., $y = \text{Conv2d}(x) + x$.
- **Unfold**: PyTorch's `torch.nn.Unfold` (im2col), which extracts sliding local blocks from the input.

Table 3: Architecture details of the autoencoder used in the proposed approach, with $\rho = \lfloor \frac{R}{2} \rfloor$ and a latent size of $16 \times 16$.

| | | Type | weight | stride | padding | Data size |
|---|---|---|---|---|---|---|
| | Encoder | Input | | | | $N \times 3 \times 32 \times 32$ |
| | | Conv2d | $3 \times 32 \times 4 \times 4$ | 2 | 1 | $N \times 32 \times 16 \times 16$ |
| | | LeakyReLU | $\alpha = 0.01$ | | | $N \times 32 \times 16 \times 16$ |
| | | Conv2d | $32 \times 64 \times 3 \times 3$ | 1 | 1 | $N \times 64 \times 16 \times 16$ |
| | | LeakyReLU | $\alpha = 0.01$ | | | $N \times 64 \times 16 \times 16$ |
| | | Conv2d | $64 \times M \times 1 \times 1$ | 1 | 0 | $N \times M \times 16 \times 16$ |
| | | ResConv2d | $M \times M \times 1 \times 1$ | 1 | 0 | $N \times M \times 16 \times 16$ |
| | | ResConv2d | $M \times M \times 1 \times 1$ | 1 | 0 | $N \times M \times 16 \times 16$ |
| | Latents | | | | | $N \times M \times 16 \times 16$ |
| | Decoder | ResConv2d | $M \times M \times 1 \times 1$ | 1 | 0 | $N \times M \times 16 \times 16$ |
| | | ResConv2d | $M \times M \times 1 \times 1$ | 1 | 0 | $N \times M \times 16 \times 16$ |
| | | ConvT2d | $M \times M \times 2 \times 2$ | 2 | 0 | $N \times M \times 16 \times 16$ |
| | | Unfold | $R \times R$ | 1 | $\lfloor \frac{R}{2} \rfloor$ | $N \times M \cdot R^2 \times 16 \times 16$ |
| | | Conv2d | $M \cdot R^2 \times 1024 \times 1 \times 1$ | 1 | 0 | $N \times 1024 \times 16 \times 16$ |
| | | LeakyReLU | $\alpha = 0.01$ | | | $N \times 1024 \times 16 \times 16$ |
| | | Conv2d | $1024 \times 1024 \times 1 \times 1$ | 2 | 0 | $N \times 1024 \times 16 \times 16$ |
| | | LeakyReLU | $\alpha = 0.01$ | | | $N \times 1024 \times 16 \times 16$ |
| | | ConvT2d | $1024 \times 3 \times 2 \times 2$ | 2 | 0 | $N \times 3 \times 32 \times 32$ |
| | Refine | Conv2d | $3 \times 32 \times 1 \times 1$ | 1 | 1 | $N \times 32 \times 32 \times 32$ |
| | | LeakyReLU | $\alpha = 0.01$ | | | $N \times 32 \times 32 \times 32$ |
| | | Conv2d | $32 \times 3 \times 1 \times 1$ | 1 | 1 | $N \times 3 \times 32 \times 32$ |
| | | Output | | | | $N \times 3 \times 32 \times 32$ |

- **ResConv2d**: A Conv2d layer with a residual skip, i.e., $y = \mathrm{Conv2d}(x) + x$.
- **Unfold**: PyTorch's `torch.nn.Unfold` (im2col), which extracts sliding local blocks from the input.

Table 4: Architecture details of the autoencoder used in the proposed approach, with $\rho = \lfloor \frac{R}{2} \rfloor$ and a latent size of $32 \times 32$.

| | | Type | weight | stride | padding | Data size |
|---|---|---|---|---|---|---|
| | Encoder | Input | | | | $N \times 3 \times 32 \times 32$ |
| | | Conv2d | $3 \times 32 \times 3 \times 3$ | 1 | 1 | $N \times 32 \times 32 \times 32$ |
| | | LeakyReLU | $\alpha = 0.01$ | | | $N \times 32 \times 32 \times 32$ |
| | | Conv2d | $32 \times 64 \times 3 \times 3$ | 1 | 1 | $N \times 64 \times 32 \times 32$ |
| | | LeakyReLU | $\alpha = 0.01$ | | | $N \times 64 \times 32 \times 32$ |
| | | Conv2d | $64 \times M \times 1 \times 1$ | 1 | 0 | $N \times M \times 32 \times 32$ |
| | | ResConv2d | $M \times M \times 1 \times 1$ | 1 | 0 | $N \times M \times 32 \times 32$ |
| | | ResConv2d | $M \times M \times 1 \times 1$ | 1 | 0 | $N \times M \times 32 \times 32$ |
| | Latents | | | | | $N \times M \times 32 \times 32$ |
| | Decoder | ResConv2d | $M \times M \times 1 \times 1$ | 1 | 0 | $N \times M \times 32 \times 32$ |
| | | ResConv2d | $M \times M \times 1 \times 1$ | 1 | 0 | $N \times M \times 32 \times 32$ |
| | | ConvT2d | $M \times M \times 2 \times 2$ | 2 | 0 | $N \times M \times 32 \times 32$ |
| | | Unfold | $R \times R$ | 1 | $\lfloor \frac{R}{2} \rfloor$ | $N \times M \cdot R^2 \times 32 \times 32$ |
| | | Conv2d | $M \cdot R^2 \times 1024 \times 1 \times 1$ | 1 | 0 | $N \times 1024 \times 32 \times 32$ |
| | | LeakyReLU | $\alpha = 0.01$ | | | $N \times 1024 \times 32 \times 32$ |
| | | Conv2d | $1024 \times 1024 \times 1 \times 1$ | 2 | 0 | $N \times 1024 \times 32 \times 32$ |
| | | LeakyReLU | $\alpha = 0.01$ | | | $N \times 1024 \times 32 \times 32$ |
| | | ConvT2d | $1024 \times 3 \times 2 \times 2$ | 2 | 0 | $N \times 3 \times 32 \times 32$ |
| | Refine | Conv2d | $3 \times 32 \times 1 \times 1$ | 1 | 1 | $N \times 32 \times 32 \times 32$ |
| | | LeakyReLU | $\alpha = 0.01$ | | | $N \times 32 \times 32 \times 32$ |
| | | Conv2d | $32 \times 3 \times 1 \times 1$ | 1 | 1 | $N \times 3 \times 32 \times 32$ |
| | | Output | | | | $N \times 3 \times 32 \times 32$ |

- **ResConv2d**: A Conv2d layer with a residual skip, i.e., $y = \mathrm{Conv2d}(x) + x$.
- **Unfold**: PyTorch's `torch.nn.Unfold` (im2col), which extracts sliding local blocks from the input.

