# OpenReview forum: "Local Autoregression with Finite-Support Random Variables for Image Generation"
_ICLR.cc/2026/Conference — ICLR 2026 Conference Withdrawn Submission_

### Official Review · Reviewer_GdiD · 2025-10-29

**Soundness:** 2
**Presentation:** 3
**Contribution:** 2
**Rating:** 4
**Confidence:** 3

**Summary:**

The paper proposes a generative model built around a standard convolutional autoencoder whose latent variables are forced into discrete ±1 values. Instead of learning a neural network to model the latent distribution (like PixelCNN in VQ-VAE), the authors explicitly compute local conditional probabilities between nearby latent variables and sample from them to generate new images. The idea is to replace learned priors with direct empirical statistics, claiming this “embraces overfitting” rather than fighting it. The result is an interpretable, local, and network-free generative process that produces patch-consistent but globally incoherent images.

**Strengths:**

1. **Explicit, non-neural autoregressive formulation**
   The paper presents an autoregressive mechanism that does not rely on a separate neural network. The local conditional probabilities are derived directly from the latent representations, offering a transparent and interpretable way to express local dependencies.

2. **Finite-support latent space**
   By restricting latent variables to discrete ±1 values, the model defines a clear and bounded generative space. This finite representation is conceptually neat and mathematically well-defined, even if not computationally scalable.

3. **Architectural control of locality**
   The idea of determining the dependency range explicitly through the decoder’s receptive field provides a clean design handle on locality. This reinforces an interpretable link between architectural structure and dependency modeling.

4. **Conceptual clarity and simplicity**
   The overall method is straightforward to implement and analyze. It avoids heavy modeling components and can be reproduced with minimal machinery, which makes it a useful conceptual testbed for exploring ideas about local generation and discrete latent representations.

**Weaknesses:**

1. **No empirical validation of the main claim**
   The central observation that pixel dependencies change after reconstruction is not supported by quantitative evidence or visualization. There are no correlation maps, covariance comparisons, or statistical measures illustrating this effect. The claim remains verbal rather than demonstrated.

2. **Unclear whether the stated problem actually exists**
   The paper builds on the premise that "overfitting" is a fundamental problem in generative modeling. This premise itself deserves scrutiny. Is it really a problem? In practice, I am not aware of generative models replicating the exact training dataset entirely or even close to that. What could anecdotally look as "overfitting" in this context is usually mode collapse, where the modes are close to exiting data instances. If the authors believe that true overfitting in the sense of exact dataset memorization occurs and is problematic, they should present empirical evidence, both at scale and in controlled minimal examples. Finally, the use of the term "overfitting" is borrowed from supervised learning and lacks a clear, rigorous definition in the context of generative modeling.

3. **Undefined and inconsistent notion of dependency**
   The notion of "pixel dependency" is used throughout the paper without definition. It remains unclear whether it refers to correlation, mutual information, or graphical conditional independence. Moreover, the argument that dependence on latent variables implies independence between pixels is conceptually incorrect, since dependency mediated through shared latent variables still constitutes dependence.

4. **Lack of explanation for the reconstruction–correlation paradox**
   The paper reports that reconstructed images exhibit different pixel correlations despite negligible reconstruction error. This is an intriguing observation but never explained. No mathematical, architectural, or empirical reasoning is provided to show how near-identical pixel values can lead to large changes in correlation. I'm not saying it's not true, I can see a way for this to happen but it's definitely not trivial and requires analysis.

5. **Unaddressed relation to VQ-VAE models**
   The proposed approach is closely related to VQ-VAE methods, which also perform latent quantization and use autoregressive priors over discrete latent codes. The paper does not acknowledge or analyze this connection, creating novelty concerns. From a methodological perspective, the work can be viewed as a simplified non-learned variant of VQ-VAE combined with empirical patch statistics.


6. **Lack of relevant baselines**
   The paper presents no comparison to a network-based autoregressive model operating on continuous latents, such as a PixelRNN-like architecture without binarization. This omission prevents understanding whether the proposed explicit local probability modeling provides any advantage over standard learned autoregressive approaches.

7. **Limited experimental validation**
   The experiments are few, small in scale, and lack ablations. Reported FID scores are not competitive with modern generative models. I see this as a minor weakness, in case that the conceptual concerns are addressed.

8. **Philosophical framing overstates the contribution**
   The contrast between "Bayesian" and "Frequentist" generative modeling is presented as a conceptual breakthrough, but operationally it amounts to removing priors and performing deterministic reconstruction. The philosophical narrative therefore adds rhetorical weight without corresponding methodological substance.

9. **That's not a VAE**
Taking a VAE and removing the sampling mechanism, even with some bottleneck regularization, makes it a regularized AE. This joins other not-well-defined and mathematically inconsistent claims throughout the paper such as "dependency" and the use of "overfitting". When things are not carefully defined they can't be judged theoretically or empirically. There are no rights or wrongs and everything feels vague.

**Questions:**

1. **About the latent structure**
   Since the decoder defines a strict local dependency graph, have you inspected whether certain latent bits become systematically correlated across distant spatial locations, effectively reintroducing long-range structure despite the local design?

2. **About the role of binarization**
   Did you experiment with intermediate quantization levels (e.g., ternary or small discrete sets) to see whether the discreteness itself or the boundedness is what matters most for the generative behavior?

3. **About the empirical probability tables**
   How stable are the locally estimated conditional probabilities across different datasets or training runs? In other words, is there evidence that these empirical local rules capture something intrinsic about natural images rather than just dataset-specific patch statistics?

---

### Official Review · Reviewer_oYnR · 2025-10-30

**Soundness:** 1
**Presentation:** 1
**Contribution:** 1
**Rating:** 2
**Confidence:** 4

**Summary:**

This paper proposes the Finite-Support Local Autoregressive (FS-LAR) model, a novel generative framework inspired by the Frequentist perspective. Instead of modeling a latent prior or imposing Bayesian assumptions, the method embraces the empirical distribution as the generative target. The model uses an autoencoder to reconstruct images, assuming that pixel dependencies in the reconstructed space differ from the originals and are fully governed by the decoder’s receptive field. By controlling this receptive field, the authors model pixel dependencies using locally dependent random variables with finite support (e.g., Rademacher-distributed ±1 latents). Global image generation is performed through sampling over these locally dependent variables without the use of neural networks. The paper claims that this design provides fidelity, interpretability, and certifiability, while avoiding the traditional fit–overfit tension in generative models.
Experiments on MNIST, CIFAR-10, and CelebA show that FS-LAR produces reasonable image quality with moderate FID scores, though still below diffusion and GAN-based baselines. The paper emphasizes theoretical interpretability and philosophical grounding over empirical performance.

**Strengths:**

**Philosophical originality**:
The paper takes a genuinely distinctive stance by rejecting Bayesian priors and embracing the empirical distribution directly. The argument connecting the Frequentist philosophy with generative modeling is interesting and thought-provoking.


**Interpretability and certifiability**:
The proposed finite-support latent variables and explicit sampling equations offer a potentially interpretable generative process that could, in theory, be exhaustively verified.

**Weaknesses:**

**Limited technical novelty**:
The core idea, modeling local dependencies in latent space via controlled receptive fields, resembles prior autoregressive and vector-quantization frameworks (e.g., PixelCNN, VQ-VAE, local transformers). The addition of finite-support random variables is more of a conceptual modification than a new modeling principle.


**Weak empirical performance**:
Reported FID scores on CIFAR-10 (≈23–40) are significantly worse than contemporary methods such as DDPM or GANs (<5). Visual quality also reveals limited semantic consistency, and the generated samples lack global coherence.


**Scalability concerns**:
The proposed sampling scheme is computationally heavy and impractical for high-resolution generation. Theoretical certifiability is attractive but infeasible in high-dimensional image spaces. Additionally, the sampling mechanism is not well-suited for GPU-based parallelization


**Comparison fairness and evaluation issues**:
The paper introduces nonstandard evaluation metrics (UIR, M-UIR) and applies PSNR filtering before FID computation, making comparisons to baselines somewhat inconsistent.

**Questions:**

Q1. How does FS-LAR fundamentally differ, in modeling capacity or learning dynamics, from two-stage autoregressive models such as VQ-VAE + Transformer?

Q2. Could you clarify how the finite-support latent variables improve generative diversity compared to standard discrete latent quantization?

Q3. Given the poor scalability of direct sampling, how do you envision extending FS-LAR to larger, higher-resolution datasets?

Q4. Figure 1 seems misleading. How does it convincingly demonstrate that pixel dependencies are entirely altered after reconstruction?

---

### Official Review · Reviewer_PueL · 2025-10-31

**Soundness:** 3
**Presentation:** 2
**Contribution:** 2
**Rating:** 4
**Confidence:** 5

**Summary:**

This paper introduces the Finite-Support Local Autoregressive (FS-LAR) model, a novel approach for image generation that leverages finite support random variables to capture local pixel dependencies. The method is rooted in a Frequentist perspective, emphasizing the empirical distribution over prior assumptions. By designing the decoder architecture to control the range of pixel dependencies, the model constructs locally dependent latent representations. These are extended into random variables with finite support, enabling global sampling for image generation. The approach promises fidelity, diversity, interpretability, and certifiability, addressing the fit–overfit tension prevalent in modern generative models.

**Strengths:**

1. Constructing generative models from a novel perspecctive: the idea of controlling the range of pixel dependencies through decoder architectures is interesting and the use of finite support random variables ensures that the generation process is interpretable and potentially verifiable.
2. Interpretability and Certifiability: The generated images are interpretable as reorganizations of locally dependent pixels or patches, and the finite support allows for theoretical verification of certifiability.
3. Addressing Fit–Overfit Tension: The paper thoughtfully discusses the inherent tension in generative modeling and proposes a method that could mitigate this issue, eliminating the need to prevent overfitting.

**Weaknesses:**

1. Limited Experimental Validation: While the paper presents some experiments, the validation on large-scale (e.g., ImageNet) high-resolution (256x256, 512x512) datasets and comparisons with state-of-the-art methods are limited, especially given the novelty of the approach.
2. Despite technically sound, the performance of the introduced generative paradigm is way blow than that of diffusion models and GANs in Figure.4. Also, diffusion models and GANs are leart on different ways to model the observed distrubution, considering that the evaluation metric FID captures the distributional discrepancy between generated and observed distributions, what's the strengths of the proposed methods for real-world distribution modeling given such underperformed performance?
3. Direct sampling, which is highlighted as the most effective strategy, involves computing probabilities for all random variable patches, potentially leading to increased computational cost, especially for high-resolution images.
4. Overall presentation: Figure.1 is not clear enough to illustrate how the pixel dependencies have completely changed. Moreover, related references should be included in Introduction and Related Works in Section.3 should be orginized in subsections.

**Questions:**

Current version needs to include more comprehensive experiments on diverse datasets, especially higher resolution images instead of CIFAR and Mnist, to better demonstrate the model's capabilities and limitations.

---

### Official Review · Reviewer_CrGy · 2025-11-03

**Soundness:** 2
**Presentation:** 1
**Contribution:** 2
**Rating:** 2
**Confidence:** 2

**Summary:**

*Note: After spending more than 4 hours on this paper, I still don’t fully understand the model’s description. Here is what I gathered:*

- This paper proposes a new image generation model called **Finite-Support Local Autoregressive (FS-LAR)**.
- The model operates on a learned binary (+1, -1) 2D latent space of an autoencoder, which must be trained beforehand.
- The sampling process is **autoregressive**, generating one latent at a time in a predefined scan order.
- Each latent is conditioned on a **limited context window** (similar to PixelCNN) to prevent overfitting.
    - To introduce stochasticity, the context binary latents are turned into a **Rademacher distribution** with added noise.
    - The sampling of the next token can be derived **analytically**—no learned generative model is required (though it’s possible to use one).
- At a high level, the method first learns the binary latents of images, then adds noise and limits the context window to sample novel combinations of latents.
- Since each latent is binary and the total number of latents is finite, the model can only generate a **finite number of images.**

    Moreover, due to the binary nature, the decoded outputs are effectively **recombinations of image patches** from the dataset.

- FS-LAR is proposed with two main properties:
    - It naturally avoids overfitting due to the limited context window.
    - It produces interpretable outputs, being essentially recombinations of dataset pixels and with limited combinations.
- The **generation quality**, however, is very preliminary.

**Strengths:**

- I like the idea of generating images by **stitching existing patches** making the model closest to non-parametric as possible. Models in this category can help us probe the question: *“What does it take to generate convincing images?”*
    - It’s worth noting that similar concepts appear in **analytical diffusion models** that require no learned model and simply recombine pixels—for example, Kamb (2024).
- I am not fully qualified to judge whether the model’s design and construction are technically sound, so I defer this part to more experienced reviewers.

**Weaknesses:**

- From a lay reader’s perspective, the model is **very hard to follow**—I did not fully understand it even after spending over 4 hours on the paper.
    - The paper is missing a clear **conceptual figure** illustrating the model architecture. Figure 1 is currently underutilized and could be redesigned to clarify the overall structure.
    - The writing style is mathematical but not always **precise**. For example, the definition of the context window (its shape, for instance) is unclear in the main text and only becomes more understandable from Table 1.

        Variables like *Z* and $\tilde{G}$ are hard to interpret without shape or datatype annotations. The reuse of *N* and *M* in different contexts (Eq. 1 and Table 1) also causes confusion.

        Many such examples make comprehension difficult.

- The **generation quality** is poor throughout the paper. A recurring issue in the “difference maps” (e.g., Figure 5) shows a **left–right split** between memorization and generalization: generated images often appear as if half the image is memorized (looks good) while the other half is unrealistic.
    - It’s unclear why this happens—possibly due to a **drift problem**, where sampled latent combinations fall too far out of distribution for the decoder to reconstruct well.
        - If this is the case, is it really possible to fix this problem in a meaningful way?
    - This also implies **low diversity**, as half of each image is essentially memorized content.
    - The use of **FID** is questionable here. The claim that “the FID of FS-LAR 35 still outperforms NCSN” (Line 408) isn’t meaningful when the generated images visibly look like two stitched halves.
    - The results don’t need to be state-of-the-art, but they should demonstrate **promise or a path to improvement**—for example, showing that performance improves with larger datasets (e.g., *N* images → good, 10*N* images → better).

**Questions:**

- What is the cause of the two-half split in the generation?
- Is this a problem where sampled latent combinations fall too far out of distribution for the decoder to reconstruct well?
- Is there way to meaningfully fix this problem? How?

---

### Note · Authors · 2026-01-26

I have read and agree with the venue's withdrawal policy on behalf of myself and my co-authors.

---

### Meta-Review · Area_Chair_ud3v · 2025-12-21

**Summary:**

This paper introduces a non-neural, finite-support local autoregressive generator over discrete autoencoder latents, emphasizing interpretability and “frequentist” sampling. While conceptually interesting, the paper is hard to understand, weakly substantiated, and empirically unconvincing.

Pros
* Interpretable, explicit local sampling without a learned prior
* Clean architectural control of locality; finite-support latents enable transparency

Cons
* Poor clarity and definitions; missing a clear end-to-end figure
* Core claims (changed pixel dependencies after reconstruction) not demonstrated
* Limited novelty vs. discrete-latent AR methods (e.g., VQ-VAE/PixelCNN)
* Weak results with visible artifacts and questionable evaluation; poor scalability

Overall, interesting idea but insufficient clarity, evidence, and empirical strength for acceptance.

**Reviewer Scores:**

n/a

---

### Decision · Program_Chairs · 2026-01-26

Reject